# CONTROL-AUGMENTED AUTOREGRESSIVE DIFFUSION FOR DATA ASSIMILATION

## ABSTRACT

Despite recent advances in test-time scaling and finetuning of diffusion models, guidance in Auto-Regressive Diffusion Models (ARDMs) remains underexplored. We introduce an amortized framework that augments pretrained ARDMs with a lightweight *controller* network, trained offline by previewing future ARDM rollouts and learning stepwise controls that anticipate upcoming observations under a terminal cost objective. We evaluate this framework in the context of data assimilation (DA) for chaotic spatiotemporal partial differential equations (PDEs), a setting where existing methods are often computationally prohibitive and prone to forecast drift under sparse observations. Our approach reduces DA inference to a single forward rollout with on-the-fly corrections, avoiding expensive adjoint computations and/or optimizations during inference. We demonstrate that our method consistently outperforms four state-of-the-art baselines in stability, accuracy, and physical fidelity across two canonical PDEs and six observation regimes. We will release code and checkpoints publicly.

## 1 INTRODUCTION

Recently, much progress has been made on inference-time scaling (Uehara et al., 2025) and finetuning (Uehara et al., 2024; Domingo-Enrich et al., 2025) in diffusion models. In parallel, Auto-Regressive Diffusion Models (ARDMs) (Ge et al., 2022; Yang et al., 2023; Yu et al., 2023; Huang et al., 2025) have emerged as powerful modeling paradigm for high-dimensional spatiotemporal dynamics in scientific applications (Pathak et al., 2024; Mardani et al., 2025). However, a traditional way of training ARDMs relies on teacher forcing (Williams & Zipser, 1989) which can lead to error acumulation over inference time rollouts. Therefore, a fundamental question is: *How can we tame rollout errors in ARDMs in a principled and computationally efficient way*.

We explore this question through the lens of data assimilation (DA). DA aims to forecast high-dimensional dynamics—such as PDE states or global atmospheric weather states—while continuously incorporating observational data from sensors, weather stations, or satellites. Without such adaptation, forecasts quickly diverge from the true trajectory, even under nearly perfect initial conditions, due to chaos in the underlying dynamics (Kalnay, 2002; Carrassi et al., 2018; Evensen et al., 2022). Classical DA schemes, such as ensemble Kalman filters and variational methods (Le Dimet & Talagrand, 1986; Courtier et al., 1998; Tr'emolet, 2006) have been enormously successful in operational weather prediction, but require costly adjoints or large ensembles (Wang & Yu, 2021) and are not straightforward to combine with learned, non-Gaussian surrogates such as ARDMs.

Mathematically, one can cast DA as a sequential inverse problem (Sanz-Alonso et al., 2023), where the goal is to adjust the generative process so that forecasts remain consistent with lossy, partial observations. In this context, diffusion moddels have emerged as powerful priors for solving inverse problems (Chung et al., 2022; Pandey et al., 2025) and can thus serve as a promising alternative for DA either at test time (Rozet & Louppe, 2023; Qu et al., 2024; Manshausen et al., 2025) or by conditioning diffusion model training directly on observations (Huang et al., 2024). However, inference-only guidance in diffusion models can be expensive due to per-instance optimization, while naïve conditional training on observations can destabilize learning and is expensive to re-train for each new observation regime. This raises a natural question: *Given a pretrained ARDM model of the dynamics, how can we finetune it to generate high fidelity forecasts which adhere to the incoming observations, without resorting to expensive per-instance optimization routines at test time?*

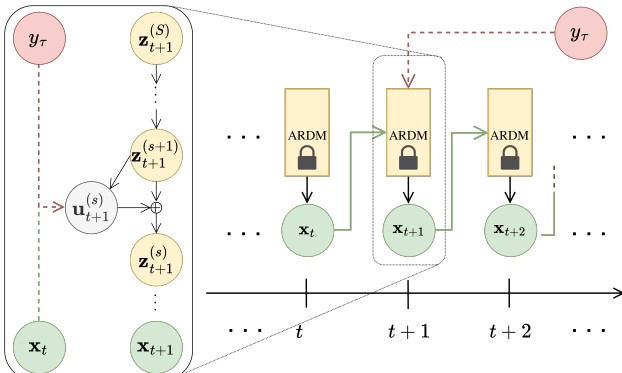

Figure 1: **Overview of (CADA)**. A pretrained autoregressive diffusion model (ARDM) generates forecasts by conditioning on previous states $x_t$ in its denoising steps. A separately trained lightweight controller injects additive controls $u_{t+1}^{(s)}$ into the denoising sub-steps $z_{t+1}^{(s+1)} \to z_{t+1}^{(s)}$, using past states and previewed future observations $y_\tau$ within the forecast window. These controls guide the frozen ARDM so that its predictions $x_{t+1}$ remain consistent with incoming observations at test time.

**Method Overview.** In this work, we build on recent formulations of diffusion guidance as stochastic control (Pandey et al., 2025; Rout et al., 2025) and propose a finetuning framework for ARDMs that embeds a *learned controller* directly into the generative dynamics. More specifically, a learned *controller* network—finetuned in a separate stage—injects affine *controls* into each denoising step of a pretrained autoregressive diffusion model, steering the dynamics to satisfy a terminal cost—which ensures that the generated forecasts align with incoming observations in a limited preview window— while satisfying a regularization penalty which enforces closeness to the unguided dynamics. At inference time, our system performs causal, feed-forward rollouts with sliding preview windows, avoiding any additional optimization or gradient-based guidance during assimilation and is thus extremely fast and stable. We study this framework on canonical chaotic PDE benchmarks and a compact ERA5-style experiment, using them as testbeds for diffusion-based DA with realistic observation sparsity. To summarize,

- We propose a diffusion-based DA framework that introduces a learned control mechanism for steering the dynamics of a pretrained ARDM to align with incoming observations (Fig. 1).

- We train the controller offline on synthetic assimilation scenarios. At test-time, the resulting *controlled ARDM* performs fully causal and feedforward rollouts without requiring any additional optimization; thus enabling fast, accurate and stable assimilation.

- Empirically, our method outperforms existing diffusion-based DA baselines on PDE and compact ERA5-based weather forecasting setups while better preserving domain-standard physical diagnostics across short and long horizons and offering upto 10x faster inference.

## 2 CONTROL AUGMENTED DATA ASSIMILATION (CADA)

### 2.1 PROBLEM STATEMENT: CHAOTIC FORECASTING WITH DELAYED, SPARSE OBSERVATIONS

When forecasting physical systems with autoregressive models, chaotic dynamics often cause small initial errors to grow rapidly, leading to instability. Real-time simulations are therefore commonly stabilized using incoming observations, a process known as data assimilation (DA). *In this DA setting, our aim is to guide a pretrained autoregressive model to produce forecasts that remain consistent with partial observations while limiting long-horizon drift.*

In this paper, we consider a sequential prediction problem involving (physical) time indices $t \in \mathbb{N}$ and a corresponding state space $\mathbf{x}_t$. For a subset of time indices $\mathcal{T} \subseteq \mathbb{N}$, we assume that we have additional observations $\mathbf{y}_\mathcal{T} \triangleq \{\mathbf{y}_\tau\}_{\tau \in \mathcal{T}}$. In a DA setup for weather forecasting, these could be

satellite data or data coming from weather stations. Importantly, we assume that these observations are noisy or incomplete and on their own insufficient to predict the ground truth state $\mathbf{x}_t$. Furthermore, we assume that we have a given, pretrained autoregressive forecasting model,

$$\mathcal{Q} = q_0(\mathbf{x}_0) \prod_{t \geq 0} q(\mathbf{x}_{t+1} \mid \mathbf{x}_t). \tag{1}$$

We will later assume this to be a conditional diffusion model, but defer details to the next subsection. Our goal will be to incorporate the additional observations $\mathbf{y}_{\mathcal{T}} \triangleq \{\mathbf{y}_\tau\}_{\tau \in \mathcal{T}}$ in order to get a better forecast *without having to retrain or even modify $\mathcal{Q}$*.

In order to optimally integrate the $\mathbf{y}-$variables, we need to assume a cost function $\Phi(\mathbf{x}_t; \mathbf{y}_t)$ that measures the compatibility of state $\mathbf{x}_t$ with observation $\mathbf{y}_t$. For example, assume a common scenario in which $\mathbf{y}_t = A(\mathbf{x}_t)$, where $A$ is some lossy data degradation operator simulating the observation process (e.g, information loss due to blur, noise, or downsampling to a smaller spatial resolution). In this case, we typically choose $\Phi = ||\mathbf{y}_t - A(\mathbf{x}_t)||^2$ as the mean squared error between the observation and the degraded ground truth state $\mathbf{x}_t$ (more details on operators can be found in App. A). This is analogous to a likelihood model $p(\mathbf{y}_t|\mathbf{x}_t) \propto \exp(-\frac{1}{\beta}\Phi(\mathbf{x}_t; \mathbf{y}_t))$ that models the compatibility of $\mathbf{x}_t$ and $\mathbf{y}_t$, where $\beta$ is a temperature parameter expressing our level of error tolerance.

A rigorous way to integrate observations $\mathbf{y}_t$ into the simulation is Bayesian inference. To this end, the optimal sampling distribution is the *tilted posterior* distribution, mediating between prior belief and data evidence,

$$\mathcal{P}^* \propto \mathcal{Q} \cdot \exp\Big(-\frac{1}{\beta}\sum_{\tau \in \mathcal{T}} \Phi(\mathbf{x}_\tau; \mathbf{y}_\tau)\Big). \tag{2}$$

In most cases of interest, this posterior distribution is intractable to compute, however, notable exceptions exist in the literature. For example, classical data assimilation methods such as Kalman filtering oftentimes assume that $\mathcal{Q}$ is a Gauss-Markov model and $A(\cdot)$ is linear (Sanz-Alonso et al., 2023), in which case the Bayesian updates are tractable. However, such assumptions severely limit the expressivity (e.g., multimodality) of $\mathcal{Q}$.

Interestingly, it can be shown that $\mathcal{P}^*$ is the optimizer of the following variational problem (proof in App. B),

$$\mathcal{C}(\mathcal{P}) \triangleq \sum_{\tau \in \mathcal{T}} \mathbb{E}_{\mathbf{x}_\tau \sim \mathcal{P}} \big[\Phi(\mathbf{x}_\tau; \mathbf{y}_\tau)\big] + \beta \, D_{\mathrm{KL}}\big(\mathcal{P} \,\|\, \mathcal{Q}\big). \tag{3}$$

If $\mathcal{P}$ is optimized over a restricted set of distributions, this amounts to a variational inference problem (Zhang et al., 2018). While it is possible to sample directly from the unnormalized tilted distribution in Eq. 2 (Vargas et al., 2023; Richter & Berner, 2024), we instead directly optimize the objective in Eq. 3 by inferring the guided dynamics $\mathcal{P}$. In the remainder of this section, we elucidate on different aspects of inferring the distribution $\mathcal{P}$ by defining the form of the unguided and guided autoregressive dynamics (Sec. 2.2), learning the guided dynamics (Sec. 2.3) and some practical instantiations (Sec. 2.4).

## 2.2 DIFFUSION-BASED DYNAMICS MODELING AND CONTROL

In order to model complex temporal dependencies, we draw on conditional Auto-Regressive Diffusion Models (ARDM) (Yang et al., 2023; Rühling Cachay et al., 2023; Price et al., 2023) for modeling dynamics. To this end, in addition to physical time steps $t \in \mathbb{N}$, we introduce additional diffusion denoising sub-steps $s \in \{S-1, \ldots, 0\}$. As follows, we denote denoising indices $(s)$ by superscript indices, and physical time steps $t$ by subscript indices (as before).

As before, we consider an autoregressive process $\mathcal{Q} = p_0(\mathbf{x}_0) \prod_{t \geq 0} q(\mathbf{x}_{t+1} \mid \mathbf{x}_t)$ with $p_0(\mathbf{x}_0)$ being an initial distribution. Since we never train or modify this ARDM's neural parameters, we suppress them in our notation. We model each transition distribution $q(\mathbf{x}_{t+1} \mid \mathbf{x}_t)$ by the *marginal distributions* (end states) of a conditional diffusion model,

$$q(\mathbf{x}_{t+1} \mid \mathbf{x}_t) = \int \left[\prod_{s=0}^{S-1} q\big(\mathbf{z}_{t+1}^{(s)} \mid \mathbf{z}_{t+1}^{(s+1)}; \mathbf{x}_t\big)\right] p\big(\mathbf{z}_{t+1}^{(S)}\big) \, d\mathbf{z}_{t+1}^{(1:S)}. \tag{4}$$

where each diffusion transition distribution is parameterized as, $q(\mathbf{z}_{t+1}^{(s)}|\mathbf{z}_{t+1}^{(s+1)}, \mathbf{x}_t) = \mathcal{N}(\boldsymbol{\mu}(\mathbf{z}_{t+1}^{(s+1)}, \mathbf{x}_t, s+1), \sigma_{s+1}^2 \boldsymbol{I}_d)$ where $\boldsymbol{\mu}$ is the pretrained denoiser. The latents $\mathbf{z}_{t+1}^{(s)}$ represent

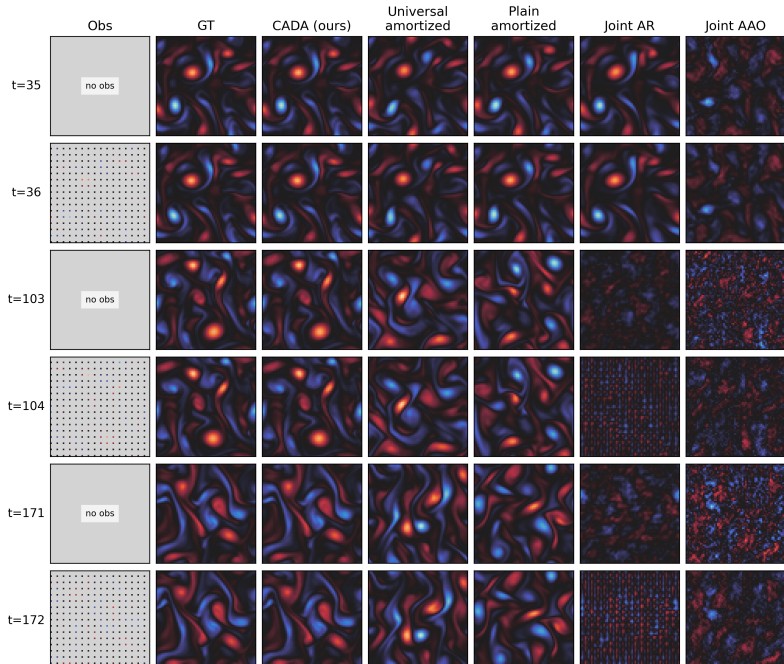

Figure 2: **Our method improves rollout stability and reconstruction consistency over long horizons.** Qualitative results on 2D Kolmogorov flow (horizon 180) show snapshots at representative timesteps, including regions with and without observations. While most baselines drift around $t \approx 35$, our approach (CADA) preserves sharp reconstructions. Joint-AR avoids catastrophic divergence but loses fine-scale structures at later steps, increasingly so for longer horizons (see Tab. 1,3).

intermediate noisy states and $p(\mathbf{z}_{t+1}^{(S)})$ represents the prior distribution for sampling initial noise. Thus, at each autoregressive timestep $t$, the next state $\mathbf{x}_{t+1} \equiv \mathbf{z}_{t+1}^{(0)}$ is obtained by simulating $S$ diffusion sampling steps conditioned on $\mathbf{x}_t$. We refer to this formulation as *unguided ARDM dynamics*.

As stated in Sec. 2.1, our goal is to steer the pretrained unguided $\mathcal{Q}$ so that the generated trajectories are aligned towards the upcoming observations while remaining close to the unguided dynamics. Therefore, we model the variational distribution $\mathcal{P}$ as an autoregressive process augmented with *control* variables $\mathbf{u}$ that help steer the process to align with incoming observations $\boldsymbol{y}$:

$$\mathcal{P} = p_0(\mathbf{x}_0) \prod_{t \geq 0} p(\mathbf{x}_{t+1} \mid \mathbf{x}_t; \boldsymbol{U}_{t+1}) \tag{5}$$

Formally, we define the transition kernel of $\mathcal{P}$ at each timestep $t$ as a conditional diffusion model,

$$p(\mathbf{x}_{t+1} \mid \mathbf{x}_t; \boldsymbol{U}_{t+1}) = \int \left[ \prod_{s=0}^{S-1} p(\mathbf{z}_{t+1}^{(s)} \mid \mathbf{z}_{t+1}^{(s+1)}; \mathbf{u}_{t+1}^{(s)}, \mathbf{x}_t) \right] p(\mathbf{z}_{t+1}^{(S)}) \, d\mathbf{z}_{t+1}^{(1:S)}. \tag{6}$$

where for brevity, we denote $\boldsymbol{U}_{t+1} = (\mathbf{u}_{t+1}^{(0)}, \dots, \mathbf{u}_{t+1}^{(S-1)})$ and $\gamma > 0$ is a scalar hyperparameter. Intuitively, at each denoising sub-step, we inject *controls* $\mathbf{u}_{t+1}^{(s)}$ which perturb the noisy state $\mathbf{z}_{t+1}^{(s+1)}$ in a direction which minimizes the training objective in Eq. 3. Furthermore, following Pandey et al. (2025), we parameterize the guided diffusion posterior in Eq. 6,

$$p(\mathbf{z}_{t+1}^{(s)} \mid \mathbf{z}_{t+1}^{(s+1)}; \mathbf{u}_{t+1}^{(s)}, \mathbf{x}_t) \triangleq \mathcal{N}(\boldsymbol{\mu}(\mathbf{z}_{t+1}^{(s+1)} + \gamma \mathbf{u}_{t+1}^{(s)}, \mathbf{x}_t, s+1), \sigma_{s+1}^2 \boldsymbol{I}_d)). \tag{7}$$

Note that $\boldsymbol{\mu}$ is the pretrained denoiser of the original ARDM $\mathcal{Q}$. We thus define our controlled process $\mathcal{P}$ in terms of an additive shift to the ARDM's noisy inputs.

Given controls $\boldsymbol{U}_{t+1}$, we can then roll-out the guided transition kernel to sample from the variational distribution $\mathcal{P}$, evaluate the rewards along the path, and backpropagate the gradients accordingly.

We refer to this parameterization as *guided ARDM*. Next, we discuss different ways of learning the controls $U_{t+1}$ given some upcoming observations within a fixed time horizon.

### 2.3 LEARNING THE CONTROLS

Given the unguided and guided ARDM formulation in Sec. 2.2, we learn the controls $U_{t+1}$ by optimizing the variational objective in Eq. 3. One alternative is to treat the controls $\{u_t^{(s)}\}$ as free variables and optimize Eq. 3 *per assimilation window* at test time, analogous to Pandey et al. (2025). This is akin to the notion of test-time scaling in diffusion models (Uehara et al., 2025). Consequently, we refer to this ablation of our method as *Test-Time Optimization based DA* (TTO-DA). Another common alternative in inverse problems is *reconstruction guidance*, where one adjusts the score at each diffusion step using $\nabla_{\mathbf{x}}\Phi(\mathbf{x}; \mathbf{y})$, without introducing a separate control policy. In the absence of a direct estimate $\mathbb{E}[\mathbf{x}_\tau^{(0)}|\mathbf{x}_t^{(s)}]$ of the future state $\mathbf{x}_\tau$, both strategies require backpropagation through the entire ARDM chain for every new forecast, which is prohibitively expensive in long autoregressive rollouts and does not naturally encode fixed-lag structure.

To mitigate these issues, we instead propose to amortize the controls by learning a controller trained over synthetic assimilation scenarios in a separate training stage. More specifically, during training, we roll out the guided ARDM (Eq. 5) trajectory and evaluate the arrival costs in Eq. 3 at observation arrival times $\tau$. Since, sampling from the guided ARDM model at any time $t$ requires specifying the controls $U_t$, we parameterize the controls at any diffusion substep $s$ and observation arrival time $\tau$ as,

$$\mathbf{u}_t^{(s)} = \boldsymbol{u}_\psi\big(\mathbf{z}_t^{(s+1)},\ \mathbf{x}_{t-1},\ \mathbf{y}_\tau,\ s,\ \tau - t\big). \tag{8}$$

where $\mathbf{z}_t^{(s+1)}$ denotes the noisy state, $\mathbf{x}_{t-1}$ denotes the forecast at the previous timestep, and $\boldsymbol{y}_\tau$ denotes a compact summary of upcoming observations. The parameters $\psi$ are then optimized, while keeping the pretrained ARDM parameters fixed. This yields an on-policy *amortized* controller that can be applied in a single forward pass at test time to correct the forecasts. We refer to the resulting framework as *Control-Augmented Data Assimilation (CADA)*.

Our proposed method CADA learns a reusable policy $\boldsymbol{u}_\psi$ from short *preview windows*: all trajectory-level optimization is done once in an offline manner. At inference, we simply run the pretrained ARDM with lightweight control corrections. Empirically, this amortized control not only reduces inference cost significantly but also yields more stable and physically faithful DA trajectories than reconstruction guidance or TTO-DA baselines (see Sec. 4).

### 2.4 PRACTICAL DESIGN CHOICES

**Anchored windows.**   During training, we operate on short, *anchored* windows of length $\Lambda$. We sample a start index $t_0$, roll out the controlled dynamics (Eq. 5) for $\Lambda$ steps, and only consider observation arrivals inside this window, i.e. $\tau \in \mathcal{T} \cap [t_0+1,\ t_0+\Lambda]$. Next, at each rollout step $t$ in this window, the controller receives a summary of the nearest upcoming observation (see App. C), lead time $s$, and the previous forecast $\mathbf{x}_t$. The controller then emits per-substep corrections as per Eq. 8 which are injected into the diffusion sub-steps via Eq. 7.

**Training on what is previewed.**   Consequently, the training objective is localized to what the controller can see. We instantiate Eq. 3 with $\beta = 0$ and a parametric path distribution $\mathcal{P}_\psi$ induced by the controlled kernel in Eq. 6. Concretely, for an anchored window $[t_0+1, t_0+\Lambda]$ with active observation times $\mathcal{A}_{t_0,\Lambda} = \mathcal{T} \cap [t_0+1, t_0+\Lambda]$, we optimize

$$\mathcal{L}(\psi) := \sum_{\tau \in \mathcal{A}_{t_0,\Lambda}} \mathbb{E}_{\mathbf{x}_{t_0+1:t_0+\Lambda} \sim \mathcal{P}_\psi(\cdot|\mathbf{x}_{t_0})} \big[\Phi(\mathbf{x}_\tau; \mathbf{y}_\tau)\big], \tag{9}$$

where the windowed rollout distribution is

$$\mathcal{P}_\psi(\mathbf{x}_{t_0+1:t_0+\Lambda} \mid \mathbf{x}_{t_0}) = \prod_{t=t_0}^{t_0+\Lambda-1} p\big(\mathbf{x}_{t+1} \mid \mathbf{x}_t; \boldsymbol{U}_{t+1}(\psi)\big), \tag{10}$$

and the per-step controls $U_{t+1}(\psi)$ are given by the policy $u_\psi$ in Eq. 8. In practice, we approximate the expectation in Eq. 9 with Monte Carlo rollouts over such windows and minimize $\mathcal{L}(\psi)$ by stochastic gradient descent. The small step size $\gamma$ and short preview horizon constrain how far each transition can deviate from the pretrained ARDM. **Algorithm 2** (App. D) summarizes the resulting training procedure on anchored windows.

**Inference by sliding previews.**    During inference, we apply the learned controller in a causal, feed-forward fashion. Starting from an initial state, we advance autoregressively and, at each step, form a preview of size $\Lambda$ over the future observation schedule, apply the controls from Eq. 8, and move forward $\Lambda$ steps before switching to the next preview. We implement this on top of a DDIM sampler (Song et al., 2020), but the preview mechanism is sampler-agnostic and only requires access to the diffusion sub-steps. We also note that inference rollouts in our experiments are substantially longer (more than 10 times) than the training windows (e.g., 60/180 steps for the Kolmogorov PDE), demonstrating that a controller trained on short previews can be reused to produce long-horizon forecasts and thus exhibits stable rollouts. **Algorithm 3** (App. D) details this sliding-preview inference scheme.

## 3    RELATED WORK

**Guidance in Diffusion Models.**    Some existing works on guidance in diffusion models rely on explicit approximations of the score of the noisy likelihood score by approximating the diffusion posterior $p\left(\mathbf{x}^{(0)}|\mathbf{x}^{(t)}\right)$ (Chung et al., 2022; Song et al., 2023; Kawar et al., 2022; Pandey et al., 2024; Pokle et al., 2024). While this can result in accurate guidance and faster sampling, a large proportion of these methods are limited to linear inverse problems, which further limits their application. More recent works (Pandey et al., 2025; Rout et al., 2025) alleviate some of these problems by formulating guidance as optimal control. Our method directly builds on top of Pandey et al. (2025) by amortizing the controls in a separate finetuning stage and extending their framework to autoregressive diffusion models. There has also been recent work in finetuning diffusion models (Clark et al., 2023; Fan et al., 2023; Domingo-Enrich et al., 2025) which is complimentary to our proposed framework.

**Data assimilation**    Several recent works utilize diffusion models for DA. Rozet & Louppe (2023) train score-based diffusion models on short trajectory segments to generate full long trajectories during inference. Their framework has been also applied in Qu et al. (2024) and Manshausen et al. (2025). However, these approaches rely on inference-time-only guidance, lacking trajectory consistency mechanisms that avoid error accumulation during observational gaps. Autoregressive methods (Shysheya et al., 2024; Gao et al., 2024) and DiffDA (Huang et al., 2024) improve stability but remain computationally expensive due to inference-time optimization. Latent space-based approaches (Foroumandi & Moradkhani, 2025; Fan et al., 2025) reduce dimensionality but introduce reconstruction biases and latent-physical decoupling errors. We address these limitations by integrating learned feedback controls directly into autoregressive diffusion denoising, enabling inference as a single forward rollout with robust long-horizon stability and substantial computational efficiency.

## 4    EXPERIMENTS

We evaluate Control–Augmented Data Assimilation (CADA) on two chaotic PDE benchmarks. To highlight challenges in delayed and sparse observations, we test six regimes combining spatial downsampling and temporal masking. Baselines include four strong diffusion-based DA methods (joint- and conditional-score) plus two CADA ablations. Our study addresses three questions: *(i) Accuracy under delayed preview*—does preview-aware control reduce drift with infrequent, lagged data? *(ii) Stability across horizons*—do improvements hold for long rollouts where chaos dominates? *(iii) Mechanism*—are gains driven by amortization or brute-force search? We next detail datasets, training, metrics, baselines, and provide comprehensive quantitative and qualitative results.

**Dataset**    The two canonical PDE benchmarks we consider are the 1D Kuramoto–Sivashinsky (KS) equation and the 2D Kolmogorov flow. Both systems exhibit nonlinear instabilities and long-range correlations, making them challenging testbeds for data assimilation with sparse observations (Du & Zaki, 2021; Lippe et al., 2023). Details on the exact PDE and data generation can be found in App E.

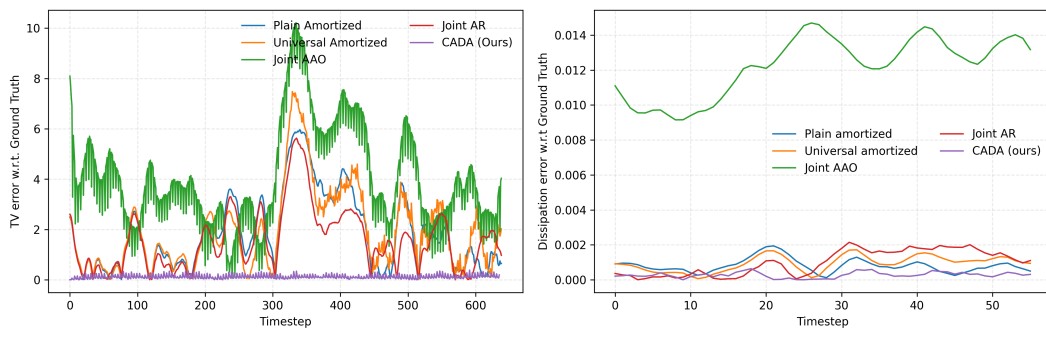

(a) Total Variation for Kuramoto-Sivashinsky
(b) Dissipation Rate for Kolmogorov Flow

Figure 3: **Our method better preserves physics-aware diagnostics under sparse observations.** (a) Total variation (TV) error for the 1D Kuramoto–Sivashinsky system over a 640-step rollout; lower TV error indicates more faithful preservation of spatial oscillations. (b) Dissipation rate error for 2D Kolmogorov flow over a 60-step rollout; accurate dissipation reflects correct energy cascade to small scales. Both figures correspond to the MR4 observation regime (details in Sec. 4).

**Observation Regimes.** We evaluate six assimilation settings for each dataset, designed to probe robustness under varying degrees of spatial sparsity and temporal delay akin to (Rozet & Louppe, 2023) and (Shysheya et al., 2024). The first set of regimes applies *spatial downsampling* with factors $\{2, 4, 8\}$ (denoted DS-2/4/8), where observations are available at every simulator step within the preview window $\Lambda$, but only on coarsened spatial grids. The second set applies *regular strided masking* with the same factors (denoted MS-2/4/8), where observations are reported every fourth simulator step within the window (temporal stride of 4) and further subsampled spatially by the given factor. In all cases, operator metadata (e.g., masks) are carried with each observation, as detailed in App. A.

**Experimental Setup.** We train a separate *controller* network for each observation regime, using a common pretrained ARDM (distinct for each dataset). We present full architectural and training details of the ARDM and controller network in App. F and App. G, respectively. The ARDM backbone in CADA employs DDIM sampling with $S=3$ denoising sub-steps per simulator transition. The preview horizon is set to $\Lambda=16$ for Kolmogorov flow and $\Lambda=54$ for Kuramoto–Sivashinsky, with the active observation selected by the nearest-arrival rule within the anchored window (see App. C). To strengthen the assimilation signal, we evaluate the arrival cost not only at observation indices but also at their intermediate denoising sub-steps, using Tweedie estimates of the forecast state at each sub-step. Forecasts are evaluated under both short and long horizons to separate near-term correction from long-term stability: 60 and 180 steps for Kolmogorov flow, and 140 and 640 steps for Kuramoto–Sivashinsky.

**Evaluation Metrics.** We evaluate forecasts using both trajectory-based and physics-aware metrics. First, we report time averaged root-mean-square error (RMSE). Second, we compute the high correlation time (HCT). Beyond pointwise errors, we assess physical fidelity by including domain-specific diagnostics. For Kuramoto-Sivashinsky, it is the *total variation* (TV) and for Kolmogorov flow, it is the *dissipation rate*. More details on these metrics can be found in App. E.

Together, RMSE and HCT evaluate assimilated forecast accuracy and temporal coherence, while TV and dissipation provide assessments of structural fidelity in chaotic PDE dynamics.

**Baselines** Score-based Data Assimilation (Rozet & Louppe, 2023) learns local joint scores over short segments under a $k$-order Markov factorization and reconstructs a full-trajectory score to sample *all-at-once* (AAO). This provides a principled joint generative model over trajectories, but is memory-intensive for long sequences and in practice requires sequential evaluation of local windows; AAO often benefits from additional corrector steps to enforce start–end consistency.

Using the same local joint score, Shysheya et al. (2024) adopt an autoregressive rollout that generates $P$ future states conditioned on the past $C$ (with $P+C=2k+1$), iterating along the horizon. Relative

to AAO, the AR factorization increases the effective Markov order seen at each step and empirically yields more stable long-range rollouts at the cost of additional neural function evaluations. Both of these joint-score based methods are referred to as *Joint AAO* and *Joint AR* in this text.

We also evaluate two conditioning architectures from Shysheya et al. (2024): *Plain Amortized*, which fixes the number of conditioning frames $C$ during training, and *Universal Amortized*, which samples $C \sim \mathrm{U}(0, \ldots, 2k)$ and uses masking to admit variable $(C, P)$ at test time while keeping a fixed input window. Both can be combined with reconstruction guidance for partial observations, following the conditional score formulations.

**Our ablations.** (i) *TTO-DA* (test-time optimization): a non-amortized variant that updates controls per step via inner optimization under a terminal-cost objective, requiring explicit rollouts to evaluate arrival costs; this parallels Pandey et al. (2025) by extending it to autoregressive temporal setting. However, unlike Pandey et al. (2025), the temporal DA problem setting doesn't give direct access to the estimate of future state $\mathbb{E}[\mathbf{x}_\tau^{(0)} | \mathbf{x}_t^{(s)}]$, necessitating a full autoregresive rollout till $\tau$ at each denoising substep to optimize controls for that substep making TTO-DA extremely expensive. (*Note that this precisely where our method of amortizing the controls via $\mathbf{u}_\psi$ and training it offline comes in handy and makes the inference extremely cheap.*) (ii) *BoN* (Best-of-$n$): a selection baseline that samples $n=16$ independent latent seeds and returns the trajectory with the lowest terminal cost—representing a simple inference-time selection heuristic used in contemporary alignment/scaling studies (Singhal et al., 2025; Gao et al., 2023).

**Qualitative and Quantitative Analysis.** Tab. 1 reports quantitative comparisons between CADA, state-of-the-art baselines, and ablations. CADA consistently outperforms all baselines across metrics, and notably maintains nearly constant RMSE from short to long rollouts. High Correlation Time (HCT; Tab. 3, see App. H) corroborates this: correlations remain high throughout full trajectories, whereas other ARDM-based models (Plain and Universal Amortized) exhibit substantial long-horizon degradation and in some regimes even struggle at short horizons. Such instability is undesirable for operational DA, where stable forecasts over fixed-lag windows are critical.

Not using amortization (TTO-DA, the test-time–optimized control variant) leads to a clear drop in performance—modest on short rollouts but increasingly severe over long horizons. This indicates that simply tilting the distribution at arrival times is insufficient; amortizing over observations and preview windows is essential for stability. The Best-of-$n$ (BoN) heuristic, which samples multiple trajectories and selects the lowest-cost one, performs significantly worse: while BoN can improve sample quality in image generation, DA requires consistent step-by-step correction rather than ex-post trajectory selection, underscoring the importance of amortized control for sequential inverse problems.

Fig. 2 and Fig. 5 (App. I) illustrate performance under masked observation regimes for long-horizon rollouts on both KS (1D) and Kolmogorov (2D). These settings are particularly challenging, as observations are sparse in both time and space. On KS, autoregressive diffusion baselines diverge after roughly 200 steps, whereas CADA remains stable for the full 640-step horizon. Joint score–based methods maintain stability but exhibit strong visual artifacts. On Kolmogorov, ARDM baselines such as Plain and Universal Amortized start diverging shortly after step 35; Joint AR is more stable but loses fine-scale detail at later times. In contrast, CADA preserves sharper, more physically consistent structures throughout.

Fig. 3a tracks total variation in the KS system, a proxy for fine-scale spatiotemporal variability. All baselines either under- or over-estimate fine scales, whereas CADA tracks the ground truth closely across the entire horizon, with small variance and minimal drift. Fig. 3b reports dissipation rate in Kolmogorov flow, a canonical diagnostic of turbulent energy transfer. Joint-score methods underestimate dissipation, and amortized ARDMs overshoot, leading to unphysical rollouts. CADA, by contrast, aligns closely with the ground truth, preserving the correct energy balance. Together, these results show that control augmentation not only reduces forecast drift but also better preserves domain-relevant physical invariants.

In addition to accuracy and stability, CADA is also substantially more efficient at inference. Tab. 2 reports wall-clock time for generating 8 trajectories in the MS-8 regime (horizon 60). CADA achieves a runtime of 6.3 s, compared to 63.4–65.6 s for conditional ARDM baselines (Plain and Universal Amortized) and 326.8–756.3 s for joint-score DA methods (Joint AAO/AR), i.e., over $10\times$ and $50$–$120\times$ speedups, respectively. This reflects that CADA performs a single forward rollout of

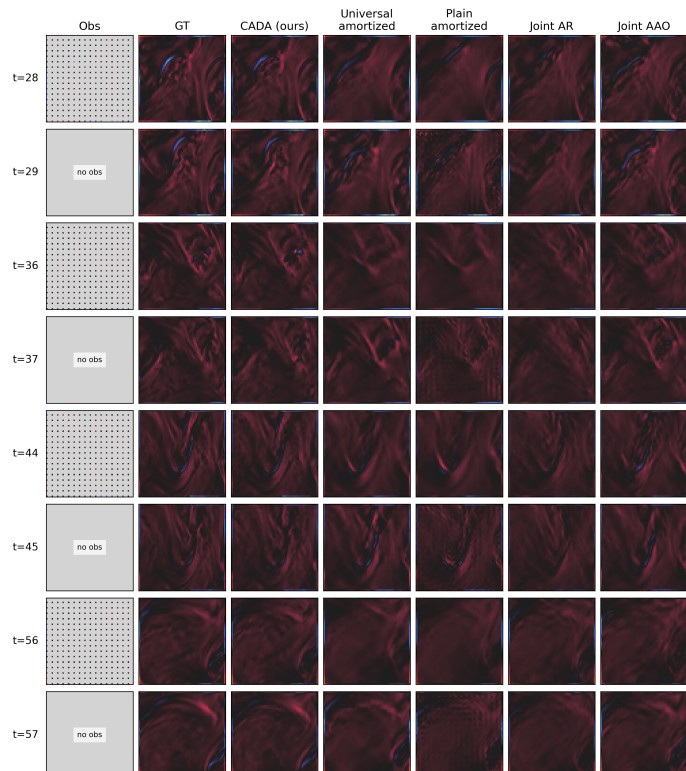

Figure 4: **ERA5 vorticity** assimilation under sparse MS-4 observations (500 hPa) (ERA5 temperature in Fig. 6). We evaluate the ability maintain fine-scale rotational structure. Rows correspond to evaluation times, with observation availability shown on the left. The ground truth (GT) reveals sharp, filamentary vorticity patterns that are notoriously difficult to preserve under sparse DA. Joint-score baselines (Joint AR/AAO) maintain broad flow patterns but lose small-scale filaments; Plain and Universal Amortized models show substantial smoothing and mode collapse. CADA uniquely retains coherent eddies and streaks even when observations are missing, demonstrating stable, physically consistent assimilation across long horizons. (see Tab. 4).

the pretrained ARDM with a lightweight controller, whereas guidance-only and joint-score methods require repeated score or gradient evaluations during sampling.

Appendix I further corroborates these findings in three additional settings. First, an ERA5-based case study on 500 hPa winds and temperature over North America shows that the same controller architecture transfers to an NWP-style surrogate and reanalysis-like observations: in the MS-4 regime, CADA reduces RMSE by roughly 3.5–4× relative to the next-best diffusion baseline (Tab. 4, Figs. 6–4). Second, a comparison to classical baselines under matched observation operators and cadences highlights that EnKF/3DVar/4DVar deteriorate sharply under aggressive downsampling and mixed-resolution regimes, whereas CADA remains accurate (Tab. 5, Fig. 7). Third, a randomized spatial-mask experiment with irregular, non-grid-aligned observations shows that CADA attains the lowest RMSE by a wide margin (Tab. 6), demonstrating robustness to irregular observation networks without architectural changes.

## 5 CONCLUSION

We introduced Control-Augmented Data Assimilation (CADA), a finetuning framework that integrates a learned control mechanism into pretrained autoregressive diffusion models. By amortizing anticipatory corrections through preview windows, CADA transforms data assimilation into a feed-forward process that is both computationally efficient and stable across long horizons.

Table 1: **Our method outperforms baselines (RMSE ↓) across six obervation regimes**. Results on Kolmogorov flow (60/180 steps) and Kuramoto–Sivashinsky (140/640 steps) under short- and long-horizon rollouts show CADA consistently superior. Ablations confirm that removing amortization (TTO-DA) or relying on heuristic selection (BoN) substantially degrades performance. Observation regimes (Sec. 4) include downsampled (DS, every step observed) and masked (MS, observations every fourth step). Refer to Tab. 3 in App.H for HCT ↑ metric.

| | DS-2 | | DS-4 | | DS-8 | | MS-2 | | MS-4 | | MS-8 | |
|---|---|---|---|---|---|---|---|---|---|---|---|---|
| | short | long | short | long | short | long | short | long | short | long | short | long |
| **Kolmogorov** | | | | | | | | | | | | |
| **CADA (ours)** | **0.016** | **0.016** | **0.020** | **0.020** | **0.138** | 0.351 | **0.017** | **0.017** | **0.024** | **0.024** | **0.060** | **0.286** |
| Joint AAO | 0.041 | 0.045 | 0.210 | 0.189 | 0.380 | 0.244 | 0.141 | 0.171 | 0.358 | 0.559 | 0.465 | 0.523 |
| Joint AR | 0.038 | 0.031 | 0.185 | 0.115 | 0.366 | **0.218** | 0.046 | 0.129 | 0.152 | 0.261 | 0.404 | 0.574 |
| Plain Amortized | 0.109 | 0.814 | 0.229 | 1.033 | 0.712 | 1.276 | 0.245 | 0.454 | 0.302 | 0.477 | 0.316 | 0.479 |
| Universal Amortized | 0.295 | 1.398 | 1.061 | 1.566 | 1.612 | 1.766 | 0.186 | 0.397 | 0.323 | 0.469 | 0.351 | 0.483 |
| TTO-DA | 0.040 | 0.243 | 0.027 | 0.115 | 0.156 | 0.401 | 0.078 | 0.298 | 0.113 | 0.357 | 0.215 | 0.433 |
| BoN | 0.258 | 0.420 | 0.264 | 0.440 | 0.299 | 0.442 | 0.265 | 0.447 | 0.266 | 0.454 | 0.306 | 0.488 |
| **Kuramoto–Sivashinsky** | | | | | | | | | | | | |
| **CADA (ours)** | **0.006** | **0.006** | **0.006** | **0.006** | **0.009** | **0.009** | **0.006** | **0.006** | **0.011** | **0.011** | **0.011** | **0.096** |
| Joint AAO | 0.017 | 0.017 | 0.091 | 0.092 | 0.417 | 0.424 | 0.045 | 0.038 | 0.210 | 0.195 | 0.614 | 0.599 |
| Joint AR | 0.018 | 0.018 | 0.091 | 0.093 | 0.413 | 0.428 | 0.026 | 0.009 | 0.041 | 0.032 | 0.134 | 0.136 |
| Plain Amortized | 0.041 | 9.787 | 0.146 | 10.73 | 1.859 | 11.81 | 0.034 | 1.163 | 0.036 | 1.165 | 0.039 | 1.211 |
| Universal Amortized | 0.043 | 5.574 | 0.146 | 6.210 | 2.096 | 6.947 | 0.041 | 1.098 | 0.044 | 1.197 | 0.048 | 1.239 |
| TTO-DA | 0.016 | 8.288 | 0.009 | 0.053 | 0.418 | 0.634 | 0.016 | 0.298 | 0.114 | 0.363 | 0.081 | 0.580 |
| BoN | 0.046 | 1.257 | 0.046 | 1.498 | 0.048 | 3.122 | 0.045 | 1.987 | 0.046 | 1.644 | 0.049 | 2.128 |

| Method | Time (s) ↓ | × CADA ↓ |
|---|---|---|
| Plain Amortized | 63.4 | 10.1 |
| Universal Amortized | 65.6 | 10.4 |
| Joint AAO | 326.8 | 51.9 |
| Joint AR | 756.3 | 120.0 |
| TTO-DA | 230.4 | 36.6 |
| BoN | 25.8 | 4.1 |
| **CADA (ours)** | **6.3** | **1.0** |

Table 2: Inference wall-clock time (seconds) for 8 trajectories in the MR8 regime on Kolmogorov flow, horizon length 60. CADA is over $10\times$ faster than conditional ARDM baselines and $50$–$120\times$ faster than joint-score DA methods.

On two canonical chaotic PDE benchmarks, CADA consistently outperforms state-of-the-art diffusion-based DA methods, yielding more accurate forecasts, improved long-term stability, and closer adherence to domain-standard physical diagnostics. Our experiments show that amortization is key: test-time-only optimization or naive trajectory selection cannot match the robustness achieved by offline-trained control policies.

Beyond PDEs, our framework suggests a general recipe for embedding control into generative dynamics. This perspective opens avenues for extending diffusion models to other sequential inverse problems where observations are delayed, sparse, or noisy—from atmospheric science and climate modeling to robotics and scientific simulation. Future work will explore adaptive preview horizons, integration with real-world observational data, and scaling to higher-dimensional systems.

**Limitations and scope.** Our study focuses on autoregressive diffusion surrogates trained on two canonical chaotic PDEs (plus a compact ERA5-style case) with a fixed preview horizon and a separate controller per observation regime. Training the ARDM prior is computationally comparable to existing diffusion-based surrogates for PDEs and NWP, while the additional cost of learning controllers is modest; nevertheless, scaling to fully global, multi-level NWP systems will require more compute. We view extensions to meta-learned controllers, adaptive previews, and richer observational operators as natural directions for future work rather than fundamental limitations of the proposed framework.

ETHICS STATEMENT

Our work focuses on amortized guidance for autoregressive diffusion models. While we present these ideas in the context of data assimilation in PDEs, our setup could be adapted for malicious terminal costs which can lead to potential misuse. Therefore, responsible deployment, monitoring, and safeguards are critical to balance performance gains with societal risks.

REPRODUCIBILITY STATEMENT

We include proofs for all theoretical results introduced in the main text in Appendix B. We include further experimental and implementation details (including model architectures and other hyperparameter choices) in Appendix G. Our code will be made available by the time of publication.

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

## A  OBSERVATION OPERATORS

Our experiments employ linear observation operators that map the full state $\mathbf{x}$ to observed signals $\mathbf{y}_k$.

**Masked observations.**  For temporally strided or spatially sparse measurements, we define a (possibly time-varying) binary mask $\boldsymbol{M} \in \{0, 1\}^{1 \times D}$ broadcast across channels, with

$$A_{\mathrm{mask}}(\mathbf{x}) \;=\; \boldsymbol{M} \odot \mathbf{x}, \qquad \Phi^{\mathrm{mask}}(\mathbf{x}; \mathbf{y}) \;=\; \frac{\|\boldsymbol{M} \odot (\mathbf{x} - \mathbf{y})\|_2^2}{\|\boldsymbol{M}\|_1}.$$

**Downsampled observations.**  For coarse-resolution sensing, we apply average pooling $P_f$ over non-overlapping $f \times f$ blocks followed by nearest-neighbor upsampling $U_f$:

$$A_{\downarrow f}(\mathbf{x}) \;=\; U_f(P_f \mathbf{x}), \qquad \Phi^{\mathrm{ds}}(\mathbf{x}; \mathbf{y}) \;=\; \|U_f(P_f \mathbf{x}) - U_f(P_f \mathbf{y})\|_2^2.$$

These operators produce the observed signals $\mathbf{y}_\tau$ that define the arrival-time costs in Eq. 3. While we restrict to masking and downsampling here, any differentiable operator $\Phi$ could be incorporated within our framework without modification.

## B  PROOF OF THE TILTED DISTRIBUTION

We show that the optimization problem in Eq. 3 admits the tilted distribution in Eq. 2 as its optimal solution.

**Setup.**  Recall the objective

$$\mathcal{C}(\mathbf{x}) \;=\; \sum_{t \in \mathcal{T}} \mathbb{E}_{\mathbf{x}_t \sim \mathcal{P}} \Big[ \Phi_t(\mathbf{x}_t; \mathbf{y}_t) \Big] + \beta \, D_{\mathrm{KL}}(\mathcal{P} \,\|\, \mathcal{Q}),$$

where $\mathcal{P}$ is the guided ARDM distribution over trajectories, $\mathcal{Q}$ the unguided distribution, and $\Phi_t$ an arrival-time cost.

**Variational form.**  Expanding the KL divergence,

$$D_{\mathrm{KL}}(\mathcal{P} \,\|\, \mathcal{Q}) \;=\; \mathbb{E}_{\mathcal{P}} \left[ \log \frac{\mathcal{P}}{\mathcal{Q}} \right].$$

Thus the objective reads

$$\mathcal{C}(\mathcal{P}) \;=\; \mathbb{E}_{\mathcal{P}} \left[ \sum_{t \in \mathcal{T}} \Phi_t(\mathbf{x}_t; \mathbf{y}_t) + \beta \log \frac{\mathcal{P}}{\mathcal{Q}} \right].$$

**Lagrangian minimization.**  Consider minimizing $\mathcal{C}(\mathcal{P})$ over distributions $\mathcal{P}$ subject to normalization $\int \mathcal{P} = 1$. The corresponding Lagrangian is

$$\mathcal{L}(\mathcal{P}, \lambda) \;=\; \mathbb{E}_{\mathcal{P}} \left[ \sum_{t \in \mathcal{T}} \Phi_t(\mathbf{x}_t; \mathbf{y}_t) + \beta \log \frac{\mathcal{P}}{\mathcal{Q}} \right] + \lambda \left( \int \mathcal{P} - 1 \right).$$

**Stationary point.** Taking the functional derivative w.r.t. $\mathcal{P}$ gives

$$\frac{\delta \mathcal{L}}{\delta \mathcal{P}} = \sum_{t \in \mathcal{T}} \Phi_t(\mathbf{x}_t; \mathbf{y}_t) + \beta\Big(1 + \log \tfrac{\mathcal{P}}{\mathcal{Q}}\Big) + \lambda.$$

Setting this derivative to zero yields

$$\log \mathcal{P} = \log \mathcal{Q} - \tfrac{1}{\beta}\sum_{t \in \mathcal{T}} \Phi_t(\mathbf{x}_t; \mathbf{y}_t) - \tfrac{\lambda+\beta}{\beta}.$$

**Closed form.** Exponentiating both sides gives

$$\mathcal{P}^*(\mathbf{x}_{0:T}) \propto \mathcal{Q}(\mathbf{x}_{0:T}) \exp\bigg(-\tfrac{1}{\beta}\sum_{t \in \mathcal{T}} \Phi_t(\mathbf{x}_t; \mathbf{y}_t)\bigg),$$

which is exactly the tilted distribution in Eq. 2.

## C  ACTIVE OBSERVATION SELECTOR

We maintain a preview buffer containing all observations from $\mathcal{T}$ that lie within a fixed lookahead horizon $\Lambda$ from index $t_0$. Each entry in the buffer is represented as a triplet $(\boldsymbol{y}_j, \boldsymbol{M}_j, \Delta_j)$, where (i) $j \in \mathcal{T}$ is the physical time index of the observation; (ii) $\boldsymbol{y}_j$ is the observed signal, lifted to full resolution when necessary; (iii) $\boldsymbol{M}_j$ is an auxiliary mask (see App. A), while for other operators $\boldsymbol{M}_j$ may be ignored or replaced with suitable metadata; and (iv) $\Delta_j = j - t + 1$ is the lead time relative to the current forecast step $t$.

At each forecast step $t$, the active preview is defined as the nearest available future observation within the lookahead window

$$\mathcal{W}_{t|t_0} \triangleq \{\, j \in \mathcal{T} : t+1 \le j \le t_0+\Lambda \,\}.$$

The chosen preview is then

$$\omega_{t|t_0} = (\boldsymbol{y}_{j^\star}, \boldsymbol{M}_{j^\star}, \Delta_{t,j^\star}), \qquad j^\star = \arg\min_{j \in \mathcal{W}_{t|t_0}} \{\Delta_{t,j} : \Delta_{t,j} \ge 0\}.$$

In words, at each step the selector activates the nearest previewed observation within the anchored preview wiindow, along with its associated metadata and lead time. More details on how the selector works in the training and inference process can be found in App. D.

## D  TRAINING AND SAMPLING ALGORITHM

---

**Algorithm 1** Preview-aware controlled DDIM one-step ($\mathbf{x}_t \to \mathbf{x}_{t+1}$)

---

**Input:** current state $\mathbf{x}_t$; preview $\omega_{t|t_0} = \{(\mathbf{y}_j, \boldsymbol{M}_j, \Delta_{t,j}) : j \in \mathcal{W}_{t|t_0}\}$ (see App. C); pretrained
  ARDM kernels $q$; control policy $\boldsymbol{u}_\psi(\cdot)$; step $\gamma > 0$

**Output:** next state $\mathbf{x}_{t+1}$ and (if applicable) arrival-time cost $\ell_{t+1}$

1: Sample parent latent $\mathbf{z}_{t+1}^{(S)} \sim p_S$
2: $\ell_{t+1} \leftarrow 0$
3: **for** $s = S-1, S-2, \ldots, 0$ **do**                            ▷ DDIM sub-steps
4:     $\mathbf{u}_{t+1}^{(s)} \leftarrow \boldsymbol{u}_\psi(\mathbf{x}_t, \mathbf{z}_{t+1}^{(s+1)}; \omega_{t|t_0},\, s)$
5:     $\tilde{\mathbf{z}} \leftarrow \mathbf{z}_{t+1}^{(s+1)} + \gamma\, \mathbf{u}_{t+1}^{(s)}$
6:     **if** $t+1 \in \mathcal{T}$ **then**
7:         $\ell_{t+1} \mathrel{+}= \Phi(\mathbb{E}[\mathbf{z}_{t+1}^{(0)} \mid \tilde{\mathbf{z}}]; \mathbf{y}_{t+1})$
8:     **end if**
9:     $\mathbf{z}_{t+1}^{(s)} \sim q\big(\,\cdot\,\big|\,\tilde{\mathbf{z}};\, \mathbf{x}_t\big)$
10: **end for**
11: $\mathbf{x}_{t+1} \leftarrow \mathbf{z}_{t+1}^{(0)}$
12: **if** $t+1 \in \mathcal{T}$ **then**                              ▷ Eq. 3 with $\beta = 0$
13:     $\ell_{t+1} \mathrel{+}= \Phi_{t+1}(\mathbf{x}_{t+1}; \mathbf{y}_{t+1})$
14: **end if**
15: **return** $\mathbf{x}_{t+1},\, \ell_{t+1}$

---

---

**Algorithm 2** Training the *controller* network

---

**Input:** pretrained ARDM kernel $q$; stream $\{\mathbf{y}_j\}_{j\in\mathcal{T}}$; preview horizon $\Lambda$; strength $\beta > 0$; optimizer for $\psi$; *controller* network $\boldsymbol{u}_\psi(\cdot)$; step $\gamma > 0$
**Output:** trained parameters $\psi$
 1: **repeat**
 2:   Sample rollout start $t_0$ and initial $\mathbf{x}_{t_0} \sim p_0$
 3:   $\mathcal{A}_{t_0,\Lambda} \leftarrow \mathcal{T} \cap [t_0{+}1,\, t_0{+}\Lambda], \quad \widehat{\mathcal{C}} \leftarrow 0$
 4:   **for** $t = t_0, t_0{+}1, \ldots, t_0{+}\Lambda - 1$ **do**
 5:     $\mathcal{W}_{t|t_0} \leftarrow \{\, j \in \mathcal{T} : t{+}1 \leq j \leq t_0{+}\Lambda \,\}$ ▷ anchored preview; see App. C
 6:     Build $\omega_{t|t_0}$ from $\mathcal{W}_{t|t_0}$ ▷ see App. C
 7:     $(\mathbf{x}_{t+1}, \ell_{t+1}) \leftarrow \text{CONTROLLEDSTEP}(\mathbf{x}_t, \omega_{t|t_0}, q, \boldsymbol{u}_\psi, \gamma)$ ▷ Alg. 1
 8:     $\mathbf{x}_t \leftarrow \mathbf{x}_{t+1}; \quad \widehat{\mathcal{C}} \leftarrow \widehat{\mathcal{C}} + \ell_{t+1}\, \mathbf{1}\{t{+}1 \in \mathcal{A}_{t_0,\Lambda}\}$
 9:   **end for**
10:   $\widehat{\mathcal{C}} \leftarrow \widehat{\mathcal{C}} \big/ \max\{|\mathcal{A}_{t_0,\Lambda}|,\, 1\}$ ▷ arrival normalization
11:   $\mathcal{L}(\psi) \leftarrow \widehat{\mathcal{C}}$
12:   Update $\psi$ by descending $\nabla_\psi \mathcal{L}(\psi)$
13: **until** convergence

---

**Algorithm 3** Preview-aware forecasting with $\Lambda$-chunk anchoring

---

**Input:** pretrained ARDM kernel $q_\theta$; trained $\boldsymbol{u}_\psi$; initial $\mathbf{x}_{t_0}$; forecast horizon $H$; preview horizon $\Lambda$; stream $\{\mathbf{y}_j\}_{j\in\mathcal{T}}$; step $\gamma > 0$
**Output:** forecast $\mathbf{x}_{1:H} = (\mathbf{x}_{t_0+1}, \ldots, \mathbf{x}_{t_0+H})$
 1: $C \leftarrow \lceil H/\Lambda \rceil$
 2: **for** $c = 0, 1, \ldots, C{-}1$ **do** ▷ chunk index
 3:   $t_0^{(c)} \leftarrow t_0 + c\,\Lambda$
 4:   $\Lambda_c \leftarrow \min\{\Lambda,\, H - c\,\Lambda\}$ ▷ last chunk may be shorter
 5:   **for** $t = t_0^{(c)}, \ldots, t_0^{(c)} + \Lambda_c - 1$ **do**
 6:     $\mathcal{W}_{t|t_0^{(c)}} \leftarrow \{\, j \in \mathcal{T} : t{+}1 \leq j \leq t_0^{(c)}{+}\Lambda_c \,\}$ ▷ anchored preview; see App. C
 7:     Build $\omega_{t|t_0^{(c)}}$ from $\mathcal{W}_{t|t_0^{(c)}}$ ▷ see App. C
 8:     $\mathbf{x}_{t+1} \leftarrow \text{CONTROLLEDSTEP}(\mathbf{x}_t, \omega_{t|t_0^{(c)}}, \{q_\theta^{(s)}\}, \boldsymbol{u}_\psi, \gamma).\text{STATE}$
 9:   **end for**
10:   *// autoregressive handoff: last state becomes next chunk's initial condition*
11:   $\mathbf{x}_{t_0^{(c+1)}} \leftarrow \mathbf{x}_{t_0^{(c)}+\Lambda_c}$ ▷ only if $c{+}1 < C$
12: **end for**
13: **return** $\mathbf{x}_{1:H}$

---

# E  DATA AND EVALUATION

**Dataset.**  The KS equation is a fourth-order nonlinear PDE modeling flame front instabilities and solidification dynamics, with dynamics $\partial_\tau u + u\,\partial_x u + \partial_x^2 u + \nu\,\partial_x^4 u = 0$, where $\nu > 0$ denotes the viscosity. We solve it on a periodic domain with 256 spatial points and a fixed time step $\Delta\tau = 0.2$. Training trajectories span $140\Delta\tau$, while validation and test trajectories extend to $640\Delta\tau$. The generation strategy and data splits follow Shysheya et al. (2024); Brandstetter et al. (2022).

Kolmogorov flow is a 2D variant of the incompressible Navier–Stokes equations, describing the dynamics of an incompressible fluid: $\partial_\tau u + u \cdot \nabla u - \nu\nabla^2 u + \frac{1}{\rho}\nabla p - f = 0, \nabla \cdot u = 0$, with velocity field $u$, viscosity $\nu$, density $\rho$, pressure $p$, and $f$ an external forcing term. Trajectories span 64 states for training and 180 states for test and validation, each represented on a $64 \times 64$ grid, with $\Delta\tau = 0.2$. Data generation and splits follow Shysheya et al. (2024); Rozet & Louppe (2023). Evaluation is carried out on the scalar vorticity field $\Omega = \partial_x u_y - \partial_y u_x$, which captures rotational structures.

**Evaluation Metrics.** HCT is defined as the last index $\ell_{\max}$ for which the Pearson correlation $\rho(\ell)$ between forecast and ground truth remains above a fixed threshold $\phi$ (we use $\phi = 0.9$): $\ell_{\max} = \max\{\ell : \rho(\ell) \geq \phi\}, \quad t_{\max} = \ell_{\max} \Delta t.$

To assess physical fidelity beyond pointwise errors, we include two domain-specific diagnostics. For the 1D Kuramoto–Sivashinsky system, where $z(\xi)$ denotes the state field over space $\xi$, we report the *total variation* (TV), $\mathrm{TV}(z) = \int \left|\partial_\xi z(\xi)\right| d\xi$, which quantifies spatial oscillations and the sharpness of evolving patterns. For the 2D Kolmogorov flow, $z(\xi, \eta)$ denotes the streamfunction, with $(\xi, \eta)$ the spatial coordinates. We measure the *dissipation rate*, $\varepsilon = \nu \int \|\nabla z(\xi, \eta)\|_2^2 d\xi \, d\eta$. This canonical diagnostic quantifies the rate at which kinetic energy is dissipated at small scales.

## F  PRETRAINING ARDM

**Implementation details.** We adapt the 1D and 2D diffusion implementations from `lucidrains/denoising-diffusion-pytorch`[1] into an autoregressive diffusion model (ARDM) tailored for PDE forecasting. Each ARDM transition corresponds to one-step forecasting via a DDIM sampler with $S{=}3$ denoising steps, $v$-parameterization, and a sigmoid schedule for $\alpha$.

The backbone is a residual U-Net with multi-resolution attention and learned sinusoidal time embeddings:

```
dim = 64,
dim_mults = (1, 2, 4, 8),
learned_sinusoidal_dim = 128
```

Attention layers are applied at intermediate and coarse resolutions, while residual blocks follow the standard Conv–Norm–SiLU design.

**Training configuration.** Models are trained with mixed precision (FP16) and exponential moving average (EMA). The configuration is:

```
train_batch_size = 32
train_lr = 3.2e-4
train_num_steps = 1000000
gradient_accumulate_every = 1
ema_decay = 0.995
ema_every = 10
```

## G  CONTROL NETWORK ARCHITECTURE

**Overview.** The *controller* network $\boldsymbol{u}_\psi$ produces controls $\mathbf{u}_t^{(s)}$ used at each denoising sub-step. In the rollout, we write $\mathbf{u}_{t+1}^{(s)} = \boldsymbol{u}_\psi(\cdot)$ for brevity; here we provide a detailed description of the architecture and conditioning.

**Inputs.** At each sub-step, the network receives five spatial tensors concatenated along channels: (i) the current latent $\mathbf{z}_{t+1}^{(s)}$, (ii) the previous state $\mathbf{x}_t$, (iii) the preview observation $\mathbf{y}_t^\star$, (iv) the auxiliary mask $M_t^\star$, and (v) the previous control $\mathbf{u}_{\mathrm{prev}}$. In addition, it conditions on scalar metadata: the preview lag $\Delta_t^\star$, the local frame index $\tau$ within the preview window, and the current $\log \mathrm{SNR}(s)$ from the DDIM schedule.

**Backbone encoder.** The concatenated inputs are passed through a shallow convolutional encoder with two $3{\times}3$ layers and group normalization. A downsample/upsample path provides limited multi-scale context: features are reduced to half resolution, then upsampled and fused back with the original resolution. A $1{\times}1$ fusion convolution followed by group normalization yields the encoded representation.

---

[1] https://github.com/lucidrains/denoising-diffusion-pytorch

**FiLM conditioning.** Each scalar input is normalized to $[0, 1]$ and embedded via a two-layer MLP of dimension `hid`. The three embeddings (lag, frame index, SNR) are concatenated and mapped to $(\gamma, \beta)$ coefficients through a linear layer. These coefficients modulate the encoded features in a FiLM style, feat $\mapsto$ feat $\cdot (1 + \gamma) + \beta$.

**Residual head.** A $3\times3$ convolutional head outputs the control increment $\Delta_\psi$. This is added to a normalized copy of the previous control $\mathbf{u}_{\text{prev}}$, producing $\mathbf{u}_t^{(s)} = \text{GroupNorm}(\mathbf{u}_{\text{prev}}) + \Delta_\psi$. At the first denoising step, $u_{\text{prev}}$ is set to zero.

**Configuration.** In our experiments we instantiate the control network as `hid=768`, with group normalization (8 groups), hidden dimension `hid` for the encoder, and FiLM embeddings of dimension `hid`. The architecture is lightweight relative to the ARDM UNet (App. F) but sufficiently expressive to incorporate preview information into the denoising dynamics.

**Training details.** Gradients flow only into $\psi$ (the UNet $\theta$ is frozen). We use gradient checkpointing at each UNet call and detach $\mathbf{u}_{\text{prev}}$ within a frame to avoid deep denoising-step recurrences; memory scales with the number of checkpoints.

## H  HCT METRIC

Tab. 3 reports the HCT metric for all our experiments.

Table 3: Quantitative comparison (HCT $\uparrow$) between our method and competitive baselines across six observation regimes (see Sec. 4). We evaluate both 1D and 2D PDE benchmarks—Kolmogorov (60/180 steps) and Kuramoto–Sivashinsky (140/640 steps)—under short and long horizons. Our method consistently outperforms alternatives. In ablations, removing amortization (TTO-DA) or using simple heuristic selection (BoN) leads to significant degradation.

| | DS-2 | | DS-4 | | DS-8 | | MS-2 | | MS-4 | | MS-8 | |
|---|---|---|---|---|---|---|---|---|---|---|---|---|
| | short | long | short | long | short | long | short | long | short | long | short | long |
| **Kolmogorov** | | | | | | | | | | | | |
| **CADA (ours)** | **60** | **180** | **60** | **180** | **50** | 50 | **60** | **180** | **60** | **180** | **60** | **70** |
| Joint AAO | 60 | 180 | 60 | 180 | 50 | 180 | 60 | 173 | 25 | 25 | 10 | 18 |
| Joint AR | 60 | 180 | 60 | 180 | 50 | **180** | 60 | 177 | 60 | 62 | 13 | 13 |
| Plain Amortized | 60 | 64 | 41 | 45 | 13 | 17 | 38 | 42 | 32 | 33 | 31 | 32 |
| Universal Amortized | 37 | 13 | 17 | 17 | 8 | 8 | 50 | 50 | 32 | 32 | 28 | 28 |
| TTO-DA | 60 | 73 | 60 | 180 | 37 | 40 | 60 | 60 | 48 | 40 | 27 | 25 |
| BoN | 40 | 40 | 32 | 26 | 21 | 26 | 40 | 26 | 32 | 26 | 32 | 27 |
| **Kuramoto–Sivashinsky** | | | | | | | | | | | | |
| **CADA (ours)** | **140** | **640** | **140** | **640** | **140** | **640** | **140** | **640** | **140** | **640** | **140** | **640** |
| Joint AAO | 140 | 640 | 139 | 640 | 139 | 640 | 140 | 640 | 140 | 640 | 129 | 633 |
| Joint AR | 139 | 639 | 139 | 639 | 139 | 639 | 139 | 639 | 139 | 639 | 139 | 639 |
| Plain Amortized | 140 | 211 | 140 | 147 | 76 | 56 | 140 | 253 | 140 | 264 | 140 | 242 |
| Universal Amortized | 140 | 262 | 140 | 168 | 62 | 62 | 140 | 286 | 140 | 217 | 140 | 274 |
| TTO-DA | 140 | 218 | 140 | 640 | 139 | 633 | 140 | 638 | 140 | 637 | 140 | 634 |
| BoN | 140 | 260 | 140 | 250 | 140 | 240 | 140 | 243 | 140 | 250 | 140 | 246 |

## I  ADDITIONAL RESULTS

We present some additional qualitative and quantitative results to further validate the efficacy of CADA.

To begin with, Fig.5 illustrates the superiority of our model on the 1D Kuramoto-Sivashinsky dataset under the long horizon regime, where despite being trained on $\Lambda = 54$, inference shows stability on horizon length (640) that is more than ten times larger. While other methods start diverging from the ground truth at around $t = 200$, CADA sustains the long inference process without any substantial degradation from ground truth, denoting long term stability.

Next, to directly address relevance to weather forecasting, we add a compact ERA5 case study: DA for $u/v$ wind components and temperature at a single pressure level (500 hPa) over North America (from 2006-2016). This intentionally modest setup (single level, single region) demonstrates that our

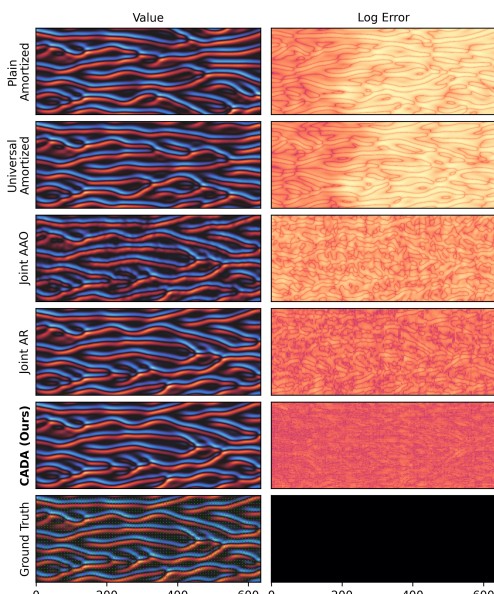

Figure 5: **Stability.** Our method yields superior long-horizon rollouts on the 1D Kuramoto–Sivashinsky PDE (horizon 640) under sparse spatiotemporal observations (green dots). Darker colors indicate lower forecast error.

| Method | RMSE ↓ |
|---|---|
| Plain Amortized | 2.1 |
| Universal Amortized | 1.9 |
| Joint AAO | 2.3 |
| Joint AR | 2.4 |
| TTo-AR | 2.3 |
| BoN | 3.6 |
| **CADA (ours)** | **0.6** |

Table 4: RMSE in the MS-4 ERA5 regime. CADA achieves substantially lower error than conditional ARDM and joint-score baselines, with a ∼3.5–4× improvement over the next-best method.

controller architecture transfers to an NWP-style surrogate and reanalysis-like observations while remaining computationally tractable. Tab. 4 reports RMSE in the MS-44 observation regime. CADA attains an RMSE of 0.6, compared to 1.9–2.4 for conditional ARDM and joint-score baselines, 2.3 for the test-time optimized control variant (TTO-DA), and 3.6 for the Best-of-$n$ heuristic. Thus, even in this ERA5-based setting, amortized control yields a roughly 3.5–4× reduction in error over the next-best method, consistent with the trends observed on the canonical PDEs benchmarks. Qualitative comparisons on ERA5 can be found in Fig. 6 and Fig. 4.

To contextualize our diffusion-based methods relative to established data assimilation techniques, we additionally evaluate a standard Ensemble Kalman Filter (EnKF) implementation under the same observation operators and cadences used in our primary experiments. As shown in Tab. 5, EnKF performs well in lightly downsampling regimes (DS-2) but its accuracy deteriorates substantially under stronger spatial downsampling (DS-8) and spatially and temporally masked settings (MS). This behavior is expected: linear–Gaussian assumptions and reliance on second-order statistics limit EnKF's ability to recover fine-scale nonlinear structures in chaotic PDE systems. By contrast, CADA maintains low RMSE across all regimes, highlighting the benefit of combining expressive diffusion surrogates with amortized control for non-Gaussian, intermittently observed dynamics. Fig. 7 details the qualitative comparison of this baseline with CADA.

Lastly, assessing robustness beyond structured downsampling and regular masking, we conduct an additional experiment using *irregular* spatial observations. Each grid point is independently revealed with probability $p = 0.125$, producing non-aligned, non-regular measurement patterns. Observations follow an irregular cadence, with a minimum separation of 2 time steps and a maximum of 6 time

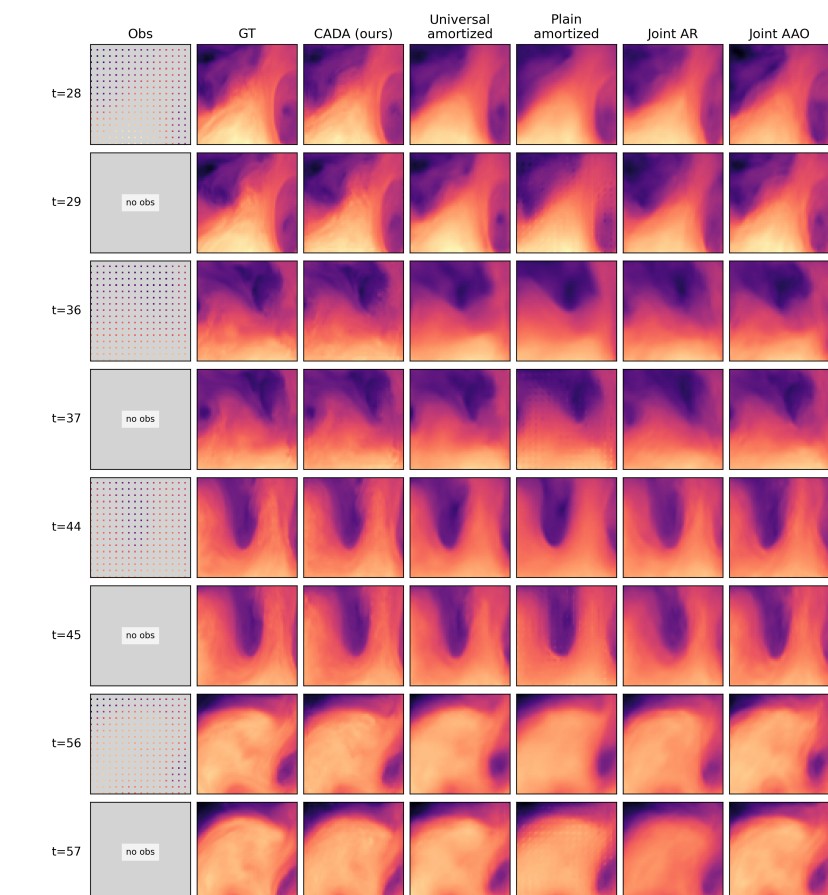

Figure 6: **ERA5 temperature** assimilation under sparse MS-4 observations (500 hPa, North America). Each row shows forecast snapshots at selected timesteps, with the leftmost column displaying the observation pattern (dense or missing). Columns compare the ground truth (GT) against CADA and four strong baselines: Universal Amortized, Plain Amortized, Joint AR, and Joint AAO. The regime is extremely challenging—observations are spatially sparse and arrive intermittently—yet CADA yields markedly sharper and more coherent temperature structures. Autoregressive diffusion baselines drift or blur fine-scale features, and joint-score models hallucinate artifacts under missing observations. CADA remains stable across all shown times, faithfully tracking the evolution of synoptic-scale fronts and gradients. (see Tab. 4).

Table 5: Classical DA baselines (EnKF, 3DVar, 4DVar) RMSE across six observation regimes for the Kolmogorov and KS benchmarks. 4DVar is competitive across both downsampling and mixed-resolution (MR) regimes. Modern diffusion-based surrogates such as CADA still significantly outperform all three classical methods across all regimes (see Tab. 1).

| Regime | EnKF ↓ | 3DVar ↓ | 4DVar ↓ |
|--------|--------|---------|---------|
| DS-2 | 0.08 | 0.08 | 0.06 |
| DS-4 | 0.09 | 0.11 | 0.18 |
| DS-8 | 0.37 | 0.09 | 0.19 |
| MR-2 | 0.31 | 0.38 | 0.26 |
| MR-4 | 0.32 | 0.41 | 0.28 |
| MR-8 | 0.35 | 0.42 | 0.33 |

steps between consecutive observations. As shown in Tab. 6, CADA attains the lowest RMSE by a wide margin (0.02), while conditional ARDM baselines experience substantial degradation. These results reinforce that preview-based amortized control remains stable even when observations deviate significantly from regular masks, and that no architectural changes are required to accommodate such irregular regimes.

Table 6: RMSE under a randomized spatial-mask regime for Kolmogorov flow. Masks select each grid point independently with probability $p = 0.125$. CADA remains robust under irregular, non-grid-aligned observations, whereas conditional ARDM baselines degrade noticeably.

| Method | RMSE ↓ |
|---|---|
| Plain Amortized | 0.28 |
| Universal Amortized | 0.26 |
| Joint AAO | 0.18 |
| Joint AR | 0.07 |
| TTO-AR | 0.13 |
| BoN | 0.32 |
| **CADA (ours)** | **0.02** |

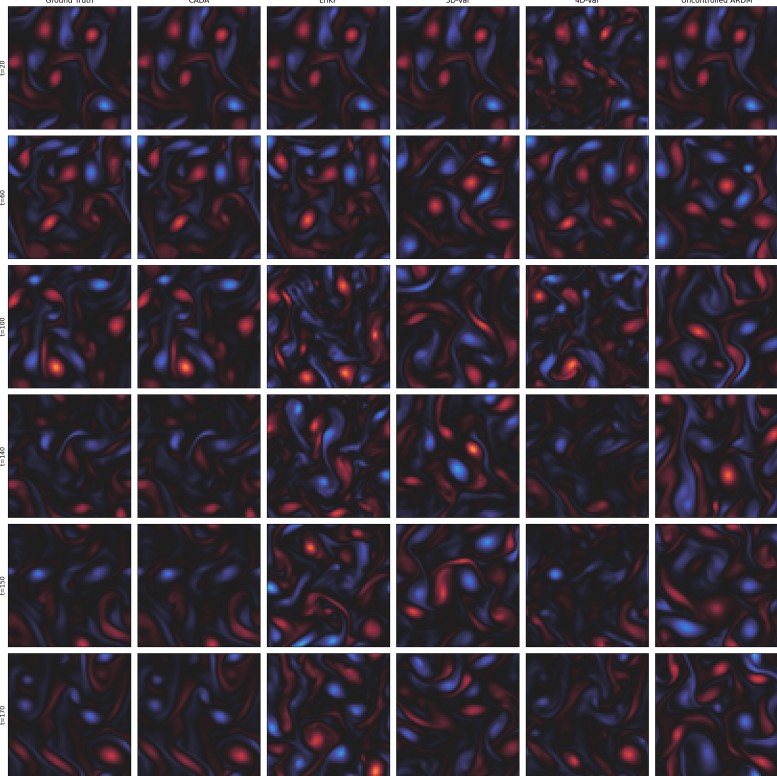

Figure 7: Kolmogorov flow assimilation under MS-4. CADA preserves coherent vortices and fine-scale filamentation across long horizons, closely matching ground truth. EnKF shows increasing phase and amplitude errors, 3D-Var and 4D-Var oversmooth or misalign small-scale structures under sparse observations, and the uncontrolled ARDM rapidly drifts. These visual trends mirror quantitative results in Tab. 5, where classical DA methods degrade significantly under mixed-resolution settings while CADA remains stable.

## J USE OF LARGE LANGUAGE MODELS

LLMs were used to assist with editing and refining the manuscript text.

