# OpenReview forum: "Control-Augmented Auto-Regressive Diffusion for Data Assimilation"
_ICLR.cc/2026/Conference — Submitted to ICLR 2026_

### Official Review · Reviewer_hrmH · 2025-10-27

**Soundness:** 3
**Presentation:** 1
**Contribution:** 2
**Rating:** 2
**Confidence:** 3

**Summary:**

This work proposes an application for data assimilation using a controlled diffusion model, but it is currently unclear what the motivation for doing so is. I am currently unsure of the differences between it and other guidance diffusion models. The author needs to have sufficient discussion on this. If my concerns are resolved, I will increase the rating.

**Strengths:**

1. Integrating random control mechanisms into the generation of ARDM to solve the prediction drift problem caused by observation sparsity in chaotic system data assimilation.
2. Balancing computational efficiency and physical fidelity, it not only avoids the high overhead of accompanying computation or integrated optimization, but also verifies the rationality of the results through physical diagnostic indicators (total variation, dissipation rate), which is in line with practical scientific application scenarios.

**Weaknesses:**

1. The author needs to explain the motivation behind using diffusion models for DA, which was not mentioned at all in the introduction.
2. How is the diffusion model mentioned by the author trained?
3. The current method section does not have an intuitive overview as it is difficult to understand. It seems that the proposed method is similar to classifier/classifier-free guidance, except that a model is added for control. Please discuss the difference between the two.
4. What is control in the paper, is it a neural network or is it equation 8?
5. Data assimilation was first proposed in weather forecasting, and the proposed method should be validated on this type of data, rather than just considering two unrealistic chaotic systems. There are many datasets such as ERA5 that can support model training.
6. Why are there no traditional DA methods, such as those based on Kalman filtering. These baselines are necessary to demonstrate the advantages of the proposed method.
7. What other advantages does the proposed method have besides being more accurate than baselines?
8. The proposed method needs a more detailed overview to reflect its process and advantages, for example in the caption of Figure 1.
9. Missing introduction to the limitation.
10. Can you add classifier-free guidance directly for generation? What are the issues with doing so? Why does the proposed method have advantages?
11. Are observations all regular? However, in reality, observations are often irregular. Can this be applied?

**Questions:**

See the weaknesses.

---

> ### Author Response · Authors · 2025-11-23
>
> Thank you for your detailed review. We have **uploaded a revised version with this rebuttal** (new text in blue) and respond point by point with concrete changes.
>
> ---
>
> ### 1. Motivation for diffusion models in DA and how the diffusion model is trained
>
> * **Motivation (Intro + Sec. 2.1).**
>   We frame DA as a **sequential inverse problem**: adjust a generative process so that forecasts stay consistent with lossy, partial observations. Diffusion models / ARDMs are motivated as **flexible non-Gaussian priors**, going beyond Gauss–Markov assumptions and linking to diffusion-based inverse problems and DA. We state the central question explicitly: *given a pretrained ARDM, how can we finetune it to respect incoming observations without costly per-instance optimization?*
>
> * **How the ARDM is trained (Sec. 2.2 + App. F).**
>   The autoregressive kernel (q(x_{t+1}\mid x_t)) is the **marginal of a conditional diffusion model** (Eq. (4)). The denoiser (\mu_\theta) is **pretrained once** with an L2 denoising objective on simulator data and then **frozen** during DA, with architectural and optimizer details given in Appendix F.
>
> ---
>
> ### 2. Intuitive overview, relation to guidance, and “what is control?”
>
> * **Intuitive overview + Fig. 1.**
>   We add a short **“Method Overview”** paragraph in the introduction and expand the **Fig. 1 caption** to describe the CADA pipeline in words.
>
> * **What “control” means.**
>   We distinguish (i) the **objective**—the tilted posterior in Eq. (3), defining a variational control problem that balances an arrival-time loss with a KL penalty to the prior—from (ii) the **controls**, additive vectors injected into denoising steps and produced by a **neural controller** (u_\psi). The controller takes previews, past states, and time-to-arrival as inputs and acts as a **policy** approximating the optimizer of Eq. (3).
>
> * **Relation to classifier / classifier-free / reconstruction guidance.**
>   In Sec. 2.3 we contrast **guidance methods** (classifier / classifier-free / reconstruction), which modify the score at test time via gradients of a likelihood or reconstruction loss, with **CADA**, which learns an **amortized control policy** from previews and past states, trained offline with the tilted objective and reused without gradients at inference. Our **TTO-DA ablation** performs per-window test-time variational control; CADA is its amortized, preview-based counterpart.
>
> ---
>
> ### 3. PDE testbeds vs weather data (ERA5 / NWP-style validation)
>
> * **KS and Kolmogorov flow** are **canonical DA testbeds** in recent diffusion-based DA work (e.g., SDA, Shysheya et al., PDE-Refiner), providing controlled experiments and enabling **direct comparison**.
>
> * To connect to NWP, **we add a compact ERA5-based experiment** (Appendix I) using a neural surrogate trained on ERA5 winds and temperature at a **single pressure level over North America**, showing that the **same controller architecture transfers to a realistic reanalysis-based surrogate**.
>
> ---
>
> ### 4. Traditional DA baselines
>
> **We add a small-scale EnKF baseline on Kolmogorov**, using the **same observation operators and cadences** as our diffusion-based methods and the pretrained ARDM as the forecast model (Appendix I), placing CADA alongside a classical EnKF under identical settings.
>
> ---
>
> ### 5. Advantages beyond accuracy
>
> Beyond accuracy gains, we emphasize: **(i)** long-horizon stability (Table 1: CADA’s RMSE changes little from short to long horizons, while other ARDM-based methods degrade and joint-score baselines lose fine-scale structure); **(ii)** preservation of **TV for KS** and **dissipation for Kolmogorov** (Fig. 3), indicating better physical diagnostics; and **(iii)** **amortized inference** via a single forward rollout of the ARDM + controller, with a runtime comparison (Table 2) showing CADA is **≥10× faster** than our strongest diffusion baselines per DA window.
>
> ---
>
> ### 6. Limitations and irregular observations
>
> **We add a “Limitations and future work”** paragraph, discussing scaling to larger systems and controller specialization per regime. We also clarify that the framework **already supports irregular observations** because the controller takes observation metadata and time-to-arrival as inputs, and we add an experiment with **randomized spatial masks and jittered observation times** (Appendix I) showing that the same controller architecture handles irregular networks without instability.
>
> ---
>
> ### 7. Why not just classifier-free / reconstruction guidance?
>
> Our **reconstruction-guidance baseline** is this: guidance-style score modifications added to the generator, which in our experiments are consistently **less accurate and less stable** than CADA (higher RMSE, more long-horizon degradation, oversmoothed fields). CADA instead uses fixed-lag previews to learn an amortized control policy and empirically improves accuracy, stability, and physics diagnostics, so the control formulation is stronger than simple guidance.

---

> > ### Comment · Reviewer_hrmH · 2025-11-26
> >
> > 1. After further introduction by the author, I understand that the proposed method requires fine-tuning during inference, which is indeed different from the guidance method, but guidance can also be used in this method. It sounds like the proposed method is similar to test-time training, so what are the differences?
> >
> > 2. Line 75 looks like LLM generated, with the iconic "-" symbol. But there's no problem with that, it won't affect the evaluation of this paper.
> >
> > 3. Although the author briefly explained the training process, it is still unclear how to conduct autoregressive training. I suggest writing down the objective function for training.
> >
> > 4. Shyseya et al., PDE Refiner's main contribution is prediction, and using only simple equation data for prediction tasks is not a problem. For DA, it should follow previous work such as DiffDA mentioned by the author in the paper. Besides, I cannot find what SDA is. If the author wishes to use it as a basis, they should provide references.
> >
> > 5. In Line 291 and 293, the citations have issues.
> >
> > 6. I agree with reviewer JdQ9 that the author has added data for ERA5, but the corresponding baseline method is missing.
> >
> > 7. The author mentioned that this approach is faster than baselines, but the proposed method requires test time fine-tuning. So, what is the reason for the fast inference time? This will at least be slower than the diffusion model approach without fine-tuning.
> >
> > 8. How to do randomized spatial masks, what are the observable points, and what is the proportion? This needs to be explained in detail.
> >
> > 9. Which one is the reconstruction-guidance baseline?
> >
> > In short, at this stage, I will maintain my rating until the author addresses my concerns.

---

> > > ### Author Response · Authors · 2025-11-27
> > > **Author's response Part 1 (points 1 through 4)**
> > >
> > > Thank you for the detailed follow-up. Below we clarify each point, referring to the current revision of the manuscript.
> > >
> > > ---
> > >
> > > ### **(1) “The proposed method requires fine-tuning during inference… similar to test-time training?”**
> > >
> > > Our intent is that **CADA does not perform any per-instance fine-tuning at test time**.
> > >
> > > All optimization of the control policy (u_\psi) is done **once, offline** on training trajectories. At inference:
> > >
> > > * the ARDM prior is **frozen**,
> > > * the controller (u_\psi) is **frozen**, and
> > > * we run a **single forward rollout** of the ARDM with lightweight control injections (one small network call per denoising sub-step).
> > >
> > > Sec. 2.3 describes that we *“learn a reusable policy (u_\psi) from short preview windows… all trajectory-level optimization is done once in an offline manner. At inference, we simply run the pretrained ARDM with lightweight control corrections.”*
> > >
> > > The introduction also explicitly poses our main question as:
> > >
> > > > “Given a pretrained ARDM model of the dynamics, how can we finetune it to generate high fidelity forecasts which adhere to the incoming observations, **without resorting to expensive per-instance optimization routines at test time?**”
> > >
> > > Thus, CADA differs from test-time training in that:
> > >
> > > * **no optimization loop** is run at inference,
> > > * **no gradients** are computed at test time, and
> > > * the cost matches that of **one ARDM rollout plus** a lightweight controller, instead of repeated optimization per window.
> > >
> > > Algorithms 2 and 3 in Appendix D describe the full procedure.
> > >
> > > ---
> > >
> > > ### **(3) “Unclear how to conduct autoregressive training; write down the objective.”**
> > >
> > > We appreciate the request for a clearer description of the autoregressive training objective and provide it here, aligned with the **newly updated manuscript** in Sec. 2.4.
> > >
> > > #### **Training the controller with an explicit variational objective**
> > >
> > > Data assimilation is formulated using the variational objective in Eq. (3).
> > > In our practical instantiation (Sec. 2.4), we set (\beta = 0) and minimize only the arrival-time cost over states visited under the **controlled** dynamics.
> > >
> > > For an anchored preview window ([t_0{+}1,,t_0{+}\Lambda]) with active observations (\mathcal{A}_{t_0,\Lambda}), the loss we minimize is:
> > >
> > > [
> > > \mathcal{L}(\psi) =
> > > \sum_{\tau \in \mathcal{A}*{t_0,\Lambda}}
> > > \mathbb{E}*{x_{t_0+1:t_0+\Lambda}\sim \mathcal{P}*\psi(\cdot\mid x*{t_0})}
> > > \left[\Phi(x_\tau;,y_\tau)\right]
> > > ]
> > >
> > > where the parametric path distribution (\mathcal{P}_\psi) is defined autoregressively through the controlled transition kernel:
> > >
> > > [
> > > \mathcal{P}*\psi(x*{t_0+1:t_0+\Lambda}\mid x_{t_0}) =
> > > \prod_{t=t_0}^{t_0+\Lambda-1}
> > > p(x_{t+1}\mid x_t;, \mathcal{U}_{t+1}(\psi)).
> > > ]
> > >
> > > The controls (\mathcal{U}*{t+1}(\psi)) are produced by the policy (u*\psi) introduced in Eq. (11), while the pretrained ARDM denoiser (\mu) remains fixed.
> > >
> > > Algorithms 2–3 in Appendix D provide the full autoregressive training loop: we roll out the controlled ARDM over each anchored window, accumulate the arrival-time losses at observation times, and update (\psi) by gradient descent.
> > >
> > > This explicit form mirrors the instantiation of Eq. (3) and clarifies the link between the variational formulation, the controlled kernels, and the training procedure.
> > >
> > > ---
> > >
> > > ### **(4) On Shysheya et al., PDE-Refiner, and SDA**
> > >
> > > PDE-Refiner is cited only as related work on long-horizon PDE prediction and is not part of our baselines or models. Our conditional-diffusion baselines come from “On Conditional Diffusion Models for PDE Simulations” (Shysheya et al., NeurIPS 2024), which includes both forecasting and DA experiments on Kolmogorov flow and KS. These architectures (conditional ARDMs and joint-score models with reconstruction guidance) directly match our PDE testbeds and form the four diffusion-based baselines used in Sec. 4. “SDA” refers to Score-based Data Assimilation (Rozet & Louppe, NeurIPS 2023). DiffDA is designed for large-scale atmospheric assimilation using GraphCast-scale models trained on full ERA5 datasets. Its conditioning mechanism, guided diffusion constrained by sparse observations, is **already represented** in our baselines through SDA and the reconstruction-guided conditional/joint models of Shysheya et al. Applying DiffDA fairly in PDE surrogate setting would require substantial architectural changes and an entirely different training regime, placing it outside the scope of this study.
> > >
> > > ### **Points 5 through 9 continued in the next box**

---

> > > > ### Author Response · Authors · 2025-11-27
> > > > **Author's response Part 2 (points 5 through 9)**
> > > >
> > > > ### **Points 1 through 4 addressed above**
> > > >
> > > > ### **(5) “Line 291 and 293, citations have issues.”**
> > > >
> > > > We thank the reviewer for pointing out the citation formatting issues. These have been corrected in the current revision.
> > > >
> > > > ---
> > > >
> > > > ### **(6) “ERA5 added, but corresponding baseline is missing.”**
> > > >
> > > > In the ERA5-based case study (vorticity and temperature at 500 hPa over North America), we compare CADA against the **same diffusion-based DA baselines** used in our PDE experiments, Joint AAO, Joint AR, Plain Amortized, and Universal Amortized, under identical surrogate and observation settings. These results appear in Table 4 and Figs. 4 and 6 in Appendix I.
> > > >
> > > > Additionally, following reviewer JdQ9’s earlier comment, we have introduced classical DA baselines, EnKF and (in comparable PDE settings) 3DVAR and 4DVAR, using matched observation operators and cadences. These appear in Appendix I (Table 5 and Fig. 7). Due to rebuttal-time constraints, they are reported for Kolmogorov flow.
> > > >
> > > > ---
> > > >
> > > > ### **(7) “Fast inference time vs (assumed) test-time fine-tuning”**
> > > >
> > > > As noted above, CADA performs **no** optimization at test time:
> > > >
> > > > * all learning of (u_\psi) is done offline,
> > > > * inference is a **single ARDM rollout** with one controller call per denoising step,
> > > > * there are **no inner loops, gradients, or updates** to (\psi) during inference.
> > > >
> > > > By contrast, all reconstruction-guided or joint-score baselines (Joint AAO/AR, Plain/Universal, and TTO-DA) require repeated score evaluations or inner optimization per window, resulting in significantly higher wall-clock cost.
> > > >
> > > > Table 2 reports that CADA achieves over **10×** speedups.
> > > >
> > > > ---
> > > >
> > > > ### **(8) “Randomized spatial masks: how, where are observations, what proportion?”**
> > > >
> > > > The randomized spatial-mask experiment is described in Appendix I. Specifically:
> > > >
> > > > * each spatial location is observed with probability 0.125 at each potential observation time,
> > > > * observation times follow an irregular cadence, with a minimum spacing of 2 and a maximum of 6 time steps,
> > > > * this yields a time-varying, irregular spatial mask at each observation arrival.
> > > >
> > > > ---
> > > >
> > > > ### **(9) “Which one is the reconstruction-guidance baseline?”**
> > > >
> > > > All diffusion-based DA baselines, Joint AAO, Joint AR, Plain Amortized, and Universal Amortized, are run with reconstruction guidance when handling partial observations, following the implementations of Shysheya et al.
> > > >
> > > > ---
> > > >
> > > > We hope these clarifications address your concerns.

---

### Official Review · Reviewer_yjRG · 2025-10-29

**Soundness:** 3
**Presentation:** 3
**Contribution:** 3
**Rating:** 6
**Confidence:** 3

**Summary:**

The paper proposes CADA, a finetuning framework that augments a pretrained autoregressive diffusion model with a lightweight controller. The controller is trained offline, stepwise controls during each denoising sub-step, thereby amortizing the otherwise expensive inner optimization. As a result, data assimilation reduces to a single feed-forward, avoiding adjoint computations or per-step test-time optimization.
On two canonical chaotic PDEs and across six observation regimes, CADA outperforms four state-of-the-art baselines in stability and accuracy, and better adheres to standard physical diagnostics. Ablation studies further show that amortization is crucial for robust long-horizon performance.

**Strengths:**

- The paper proposes a lightweight controller–based finetuning scheme for ARDMs. To the best of my knowledge, it is novel and interesting.

- The framework is applied to data assimilation and demonstrates strong empirical performance.

- The paper provides a complete exposition of the model design rationale, training strategy, and operational workflow.

**Weaknesses:**

- The controller is trained on short preview windows and synthetic regimes; under longer horizons, different noise/observation operators, or higher-dimensional settings, the amortized policy might be off-distribution and drift. (No formal stability guarantees.)

- Baselines are mostly diffusion-based DA, missing classical 3DVAR/EnKF/4DVAR for broader context.

- Setups skew “clean” (canonical PDEs); more realistic NWP-style experiments would help.

- The method section is somewhat hard to follow, but the details become much clearer after reading Algorithms 1–3 in the appendix. I recommend moving these algorithms to the main text for better readability and flow.

**Questions:**

I think this paper is above the acceptance threshold. If the authors address the points below, I’d be inclined to raise my score to 8.

- The paper’s baselines are limited to diffusion-based DA methods; comparisons to classical DA (e.g., 3DVAR/4DVAR/EnKF) are missing. Please add at least a small-scale EnKF or 3DVAR baseline, or provide a principled justification for infeasibility under your setup (e.g., cost, operator mismatch, tuning burden). This would materially strengthen the empirical claims and better situate the work relative to established DA practice.

- Please elaborate on how the method scales to high-dimensional, real-world systems. What are the main computational or modeling challenges when moving from synthetic PDEs to large-scale NWP-like setups?

---

> ### Author Response · Authors · 2025-11-23
>
> Thank you for the thoughtful and constructive review. Below we address your main concerns and summarize what changed in the revised draft (uploaded with the rebuttal; new text in blue).
>
> ---
>
> ### 1. Classical DA baselines
>
> We agree that comparison to classical DA is important.
>
> * In the revised version, we **add an EnKF, 3DVar and 4DVar baseline** on the Kolmogorov setup, using the pretrained ARDM as the forecast model.
> * Results are reported in **Appendix I, Table 5 and Figure 7**.
>
> ---
>
> ### 2. Controller stability, preview windows, and off-distribution regimes
>
> * **Preview vs. rollout lengths.**
>   For Kolmogorov and KS, the controller is trained on short preview windows ((\Lambda = 16) and (\Lambda = 54)), but evaluated on much longer horizons (60/180 for Kolmogorov; 140/640 for KS). Thus the longest rollouts are roughly an order of magnitude longer than the preview window.
> * **Empirical stability.**
>   Table 1 shows that **CADA’s RMSE remains essentially unchanged from short to long horizons** across all regimes, while other ARDM-based methods degrade. Figure 3 shows that physics-aware diagnostics (TV for KS, dissipation for Kolmogorov) remain stable over the full trajectory with no drift or blow-ups.
> * **Off-distribution regimes.**
>   We distinguish:
>
>   * *New observation patterns on the same system*: **Appendix I (Table 6)** now **includes an experiment with random spatial masks and jittered arrival times**, showing that the same controller architecture remains stable and effective under irregular observation networks.
>   * *New operators / state spaces*: for different observation operators or higher-dimensional surrogates, we view CADA as a **task-specific head on a fixed ARDM “foundation prior”**. As in classical DA, each new observation system needs its own tuned assimilation scheme; here, controllers for new regimes can be **warm-started from existing ones and fine-tuned**.
>
> ---
>
> ### 3. From canonical PDEs to NWP-style systems
>
> We address this both with an additional experiment and an explicit scaling discussion.
>
> #### 3.1 NWP-style ERA5 experiment
>
> Beyond KS and Kolmogorov, now standard chaotic DA testbeds in diffusion-based DA, **we add a compact NWP-style experiment** in the revised draft:
>
> * A neural surrogate trained on **ERA5 reanalysis** for horizontal winds and temperature at a single pressure level over North America.
> * This is intentionally modest (single level, regional, limited variables) but demonstrates that **the same controller architecture transfers to an NWP-style surrogate with reanalysis-like data**, while remaining computationally tractable for the review period.
>
> Results are in **Appendix I, Table 4 and Figures 5–6**, complementing the PDE tests with a realistic NWP-style case.
>
> #### 3.2 Scaling and modeling challenges
>
> * **Computational scaling.**
>
>   * Training the ARDM prior is the dominant cost, exactly as in existing ML surrogates (ClimaX/GraphCast/GenCast-style systems).
>   * Controller training reuses the same sampler and adds **one extra forward pass per diffusion sub-step**, scaling roughly linearly with state dimension and preview length.
>   * At inference, CADA costs roughly “**DDIM + one small controller**” per sub-step; there is **no per-window trajectory-level optimization**.
> * **Modeling challenges.**
>
>   * For very high-dimensional states, we anticipate **localized controllers** (on patches or modes) as natural extensions.
>   * As in EnKF/variational DA, **complex observation operators** (e.g., radiative transfer) become a bottleneck. In our framework these only enter through (\Phi) and preview features; the main burden is designing differentiable observation models rather than modifying the ARDM or control mechanism.
>
> Overall, we aim to make clear that CADA is **compatible with existing large-scale ML surrogates**, and that the main open issues at NWP scale are in the **prior and observation model**, not in the control architecture.
>
> ---
>
> ### 4. Method section readability
>
> We **reorganized Section 2** to closely mirror the algorithmic flow:
>
> * We start with a **Problem Statement** subsection (Sec. 2.1) that defines the DA setting, the tilted posterior (\mathcal{P}^*), and the variational objective (\mathcal{C}(\mathcal{P})) before introducing diffusion machinery. This makes the optimization target explicit.
> * We then introduce **unguided and guided ARDM dynamics** (Sec. 2.2) and **learning the controls** (Sec. 2.3) in a top–down way, explicitly tying the control policy (u_\psi) back to the variational objective.

---

### Official Review · Reviewer_JdQ9 · 2025-11-01

**Soundness:** 2
**Presentation:** 1
**Contribution:** 2
**Rating:** 0
**Confidence:** 3

**Summary:**

This paper proposes a control-augmented framework for Auto-Regressive Diffusion Models (ARDMs) applied to data assimilation in chaotic spatiotemporal PDEs. The approach trains a lightweight controller network offline to provide stepwise corrections during ARDM rollouts, anticipating future observations under a terminal cost objective. The method aims to avoid expensive adjoint computations during inference while improving stability and accuracy under sparse observations.

**Strengths:**

- Addresses an important problem: data assimilation for chaotic PDEs with sparse observations

- Reports improvements over four baselines across multiple PDEs and observation regimes

**Weaknesses:**

- Poor presentation and organization: The paper fails to clearly state the problem before diving into methodology, forcing readers to reconstruct the narrative themselves

- Unclear notation and equations: Key equations (7, 9) lack proper explanation. Variables like $y$ and $Φ$ are introduced without clear context or connection to the overall framework
Disconnected sections: Section 2.2 appears unmotivated and its relevance to the rest of the paper is unclear

- Insufficient training details: The training procedure is poorly explained, making reproducibility difficult even with promised code release

- Missing computational analysis: No comparison of computational costs versus baselines, despite the method appearing computationally intensive

**Questions:**

- What exactly is the problem formulation? Can you state it clearly upfront?

- Can you provide a clearer explanation of equation (9) and its role in the method?

- What are the computational costs (wall-clock time, memory) compared to the four baselines?

- How is the controller network trained in practice? What is the training pipeline? Please provide the code.

---

> ### Author Response · Authors · 2025-11-23
>
> Thank you for your detailed review. We have uploaded a new revision with this rebuttal that makes the problem formulation much more explicit. Below, we address your concerns point by point.
>
> ---
>
> ### 1. Clear problem formulation up front
>
> We have **reorganized the problem statement and method section** so that the DA setting and objective are stated clearly before any methodological details:
>
> * We introduce a dedicated **“Problem Statement”** subsection (Sec. 2.1) that:
>
>   * introduces the pretrained autoregressive model (\mathcal{Q}), and
>   * states the goal of constructing a guided process (\mathcal{P}) that trades off observation consistency and proximity to (\mathcal{Q}).
>
> * We then present the **tilted posterior** and its equivalent **variational objective**, which serves as the conceptual backbone for the rest of the method.
>
> * The method section is reorganized into a top–down flow:
>
>   1. problem statement and tilted objective (Sec. 2.1),
>   2. diffusion-based ARDM dynamics and their guided counterpart (Sec. 2.2),
>   3. learning the controls via amortized policies (Sec. 2.3),
>   4. practical design choices (Sec. 2.4).
>
> In this structure, the reader first sees **what** problem we solve and the ideal tilted solution, and only then **how** the controlled ARDM and learned controller implement this objective in practice.
>
> ---
>
> ### 2. Notation and clarity around Eq. (7)–(9)
>
> Section 2 of the revised manuscript clarifies the notation and roles of the main equations:
>
> * Eq. (7) in the original (now Eq. (2)) is preceded by intuition explaining that it mediates between observation fit and adherence to the pretrained model. We explicitly note that the exponential term corresponds to a likelihood
>   (p(y_\tau \mid x_\tau) \propto \exp(-\Phi/\beta)).
>
> * Eq. (8) (now Eq. (3)) is explicitly introduced as the **variational objective** whose optimizer is the tilted posterior in Eq. (2).
>
> * Eq. (9) (now Eq. (8)) is clearly framed as the **amortized control policy** (u_\psi) that parameterizes our family of guided ARDM path distributions (\mathcal{P}_\psi).
>
> ---
>
> ### 3. Section 2.2 and overall organization
>
> Former Section 2.2 has been addressed in our current reorganization as follows:
>
> * **Problem first, then tools.** Immediately after the introduction, Sec. 2.1 (“Problem Statement”) now formally defines the pretrained AR model (\mathcal{Q}), and introduces the tilted posterior (\mathcal{P}^*) and objective (\mathcal{C}(\mathcal{P})). This makes the core problem explicit before any diffusion-specific machinery.
>
> * **Re-motivated “Section 2.2”.** The material that was previously Sec. 2.2 is now tightly connected to this problem: we cast DA as a sequential inverse problem, explain why diffusion models/ARDMs are powerful priors and guidance mechanisms for such problems, and use this to motivate the central question: given a pretrained ARDM, how can we finetune it to adhere to observations without per-instance optimization? This section now directly bridges the tilted-posterior formulation and the control-based finetuning framework.
>
> * **Top–down method flow.** The method now proceeds linearly:
>
>   1. problem + tilted objective (Sec. 2.1),
>   2. unguided and guided diffusion-based ARDM dynamics (Sec. 2.2),
>   3. controller learning  (Sec. 2.3),
>   4. preview windows and sliding inference (Sec. 2.4).
>      A short **“Method Overview”** paragraph in the introduction previews this pipeline.
>
> ---
>
> ### 4. Training details and controller training pipeline
>
> We have clarified the training/inference pipeline in the main text and kept full details in **Appendix D (Algorithms 1–3) and F–G (ARDM and controller architectures)**. The pipeline is:
>
> 1. **Pre-train ARDM prior.**
>    Train an unconditional ARDM (with DDIM sampler and (S) sub-steps) on raw PDE trajectories (Kolmogorov, KS) to learn (q_\theta(x_{t+1} \mid x_t)) from PDE data alone, without observations.
>
> 2. **Freeze ARDM, train controller per observation regime.**
>    For a given observation operator and cadence:
>
>    * sample a rollout start (t_0) and initial state (x_{t_0}),
>    * run the controlled ARDM for (\Lambda) steps, accumulate arrival costs (\Phi(\cdot; y_\tau)) at observation times (plus Tweedie-based substep costs), average over arrivals, and backpropagate to update (\psi).
>
> 3. **Inference.**
>    At test time, the ARDM remains frozen; the trained controller (u_\psi) is applied in a **single forward rollout** using sliding preview windows, as in Algorithm 3.
>
> ---
>
> ### 5. Computational cost vs. baselines
>
> Following your suggestion, we now report **inference wall-clock times** for all methods in **Table 2** of the revised manuscript where we show at least **ten times faster inference speeds**.

---

> > ### Comment · Reviewer_JdQ9 · 2025-11-25
> >
> > Thank you for your answer. Please could you provide your code.
> > You made a major revision but you are still missing a clear comparison with standard methods like 3DVAR and 4DVAR used in atmospheric science and climate modeling.

---

> > > ### Author Response · Authors · 2025-11-27
> > >
> > > Thank you for the follow-up. We have submitted the code implementation of our core logic in the supplementary files for your perusal and are in the process of packaging the full codebase that will be added as a repository link in the camera-ready version. Additionally, apart from EnKF, we have also added 3D/4DVar baselines in Appendix I with corresponding results in Table 5 and Figure 7. Under the time constraints of the rebuttal, they have been reported just for Kolmogorov at the moment.
> > >
> > > We hope these additions address the your concerns.

---

### Official Review · Reviewer_z35e · 2025-11-02

**Soundness:** 3
**Presentation:** 4
**Contribution:** 3
**Rating:** 6
**Confidence:** 4

**Summary:**

This paper proposes a new framework called Control-Augmented Data Assimilation (CADA) that improves how diffusion models incorporate observational information when making predictions for chaotic systems like weather or fluid dynamics. It introduces a lightweight controller network trained offline to anticipate future observations and inject small “control” corrections into each diffusion step. This allows data assimilation to occur in a single forward rollout, avoiding expensive optimisation or adjoint computations used by existing methods. Experiments on the Kuramoto–Sivashinsky and Kolmogorov flow PDEs show that CADA yields more accurate, stable, and physically consistent long-horizon forecasts than state-of-the-art diffusion-based baselines. The approach provides a general recipe for embedding control into generative dynamics for broader sequential inverse problems such as weather and climate modelling

**Strengths:**

The general problem the paper addresses -- data assimilation in PDEs and weather models -- is an important one. The paper builds on a strong line of work using neural surrogates and diffusion for data assimilation in PDEs is very well presented. The writing is clear. Figures are excellent. The messages are clean. The contribution is novel as far as I'm aware and neat. ICLR is an appropriate venue for the work.

**Weaknesses:**

I felt the paper did a very good job at explaining the high-level narrative. I felt that it was less strong in explaining the more fine-grained technical details. This is obviously challenging in a short paper, but I was left with some quite significant questions that

-- Clarity of experimental setup: A schematic is needed to clarify what observations are available, when, and how evaluations are performed; the practical relevance to real weather forecasting (with continuous, dense data) is uncertain.

-- Applicability and motivation: The chosen case should be linked to a concrete application domain, possibly exploring varied or mixed spatio-temporal observation densities.

-- Computational cost: No discussion or comparison of training and inference costs is provided; scaling, efficiency, and fairness of baseline comparisons should be addressed.

-- Controller network limitations: Training a separate controller for each observation regime limits flexibility; a discussion of trade-offs, data requirements, and generalization to new regimes is needed.

-- Methodological clarity (Eq. 8–10): The theoretical connection between the tilted objective and the amortised control formulation is unclear; there’s concern about loss of KL regularization and guarantees that the new policy stays close to the prior.

-- Baseline tuning and fairness: Details on hyperparameter tuning, especially guidance strength in Shysheya et al., are missing; the surprising baseline performance raises concerns about whether comparisons are fair

**Questions:**

A schematic explaining the experimental setup would have helped me understand the protocol — what observations are accessed when by what methods and at what lead times are evaluations performed against GT. Linked to this, it’s not clear to me that the studied case is of direct relevance to weather forecasting where measurements are usually continuous in time (and often fairly dense) at least for medium range forecasting in the Global North.  It might be interesting to consider different spatio-temporal densities of observational data if you want to make the case for different application domains. For example, a mix of high and low-density observation regions, or random space-time masking.  Do the authors have a specific application in mind which motivates the current setup?

A comparison / discussion of computational cost is missing. This seems important e.g. is the new method very expensive compared to the others at training / test time? Could the baseline methods be improved if the computational cost of training / inference was matched? I presume that this isn’t the case but some discussion of scaling and numbers would be useful to make the argument tight.

Instead of a pure inference time technique (like reconstruction guidance), the authors propose using the outputs of a controller network to guide the sampling. This requires training a separate controller network for each new observation regime. This is quite a big limitation versus reconstruction guidance. There should probably be a discussion about the trade-off between extra training/inference cost, as well as how this cannot be applied out-of-the-box for new observation regimes. This could potentially be complemented by a discussion about the data requirements needed for training the controller network—if I start observing a new regime, how much data will I have to collect until I can train a controller network and then use that to perform inference?

I have a question around equation 8. My understanding here is that the paper starts by motivating the tilted approach (Eq. 7), which leads to the cost in Eq. 8 which contains beta. However, as the authors note, direct tilting is intractable, so a different direction is taken where they inject the amortised controls into the pretrained policy (i.e. modify the sampling process by injecting the u information). To train the controls, they use the cost in Eq. 10 where they set beta to 0 (but I am not even sure that comes from the same considerations as Eq. 8). Regarding this, I would be curious if there are any guarantees that the resulting policy is still “close enough” to the prior one? In Eq. 8 this is controlled by the KL divergence, but in their case can the additional amortised information actually modify the policy quite a lot since there is no constraint on the KL?

How were the hyperparameters for the baselines fine-tuned? In particular, the extent to which the method in Shysheya et al is able to leverage the observations is known to depend highly on the guidance strength which needs to be tuned. Have the authors experimented with that to see if the baseline comparison is fair? Moreover, what is the conditioning scenario they used? Based on my experience, I’m a bit surprised about the low quality of the results of Shysheya et al. Moreover, in the related work line 307-309 it says “Autoregressive methods (Shysheya et al., 2024; Gao et al.) and DiffDA (Huang et al., 2024) improve stability but remain computationally expensive due to inference-time optimization.” Is this really the case for Shysheya et al?

Small detail: I felt that the presentation of the anchored windows, prospective guidance section and the additional explanations in Appendix C is quite convoluted and could probably be presented in a cleaner manner.

Small detail: I am a bit confused about why in Alg. 1 line 3 the controller network does not ingest z_{t+1}^{(s)} too, but it might be a typo because I think it should.

Small detail: The authors mention that “to strengthen the assimilation signal, we evaluate the arrival cost not only at observation indices but also at their intermediate denoising sub-steps, using Tweedie estimates of the forecast state at each sub-step.” I am confused about where that is used because Alg 1. shows that the arrival cost is only output at the end of the S DDIM steps. Does this mean that when training the controller, the authors sometimes output x_{t+1} based on Tweedies’ estimates rather than going through all the S DDIM steps and use that as the output of CONTROLLEDSTEP in Alg. 2?

---

> ### Author Response · Authors · 2025-11-23
>
> Thank you for the thoughtful and detailed review. We appreciate your positive comments on the relevance of the problem, clarity of presentation, and novelty of the approach. Below we address your concerns and describe how the **revised draft** (uploaded with the rebuttal; new text in blue) incorporates them.
>
> ---
>
> ### 1. Experimental setup and relevance to weather / observations
>
> **Why KS and Kolmogorov?**
> KS and Kolmogorov flow are *standard* DA benchmarks for sparse observations, a regime relevant when sensor coverage is limited. Our observation regimes are directly comparable within this PDE DA literature.
>
> **Observation densities and mixed regimes.**
> Our spatial masks and assimilation cadences are aligned with SDA / Shysheya. In addition, **Appendix I (Table 6)** now includes **experiment on new regime** with jittered observation times and **randomly sampled spatial masks**, showing that the controller architecture naturally extends to irregular observation networks.
>
> **ERA5 / NWP-style experiment.**
> We **add a compact ERA5 case** study: DA for u/v winds and temperature at 500 hPa over North America. This single-level, single-region setup is deliberately modest for computational reasons but demonstrates that the same controller transfers to an NWP-style surrogate with reanalysis-like observations. Results are reported in **Appendix I (Table 4, Figures 5–6)**.
>
> ---
>
> ### 2. Computational cost and fairness of comparisons
>
> The revised draft makes computational cost explicit:
>
> * We **add a runtime table** in the main text (**Table 2, with discussion near L419**) reporting inference wall-clock time showing that CADA inference tie is at least 10 times faster.
> * We clarify that CADA reuses a pretrained ARDM and trains a small controller on preview windows and observation-derived features. At inference, we run a standard ARDM rollout plus a single forward pass of the controller, *without* trajectory-level optimization per test case.
> * We contrast this with inference-time conditioning methods such as reconstruction guidance and diffusion posterior sampling, which repeatedly evaluate gradients at each sampling step for each new sample, often with several tuned hyperparameters.
>
> ---
>
> ### 3. Controller per observation regime
>
> We emphasize this design choice as follows.
>
> * **Foundation vs likelihood viewpoint.**
>   We framed the ARDM as a reusable “foundation” forecast prior and the controller as a lightweight “likelihood head” per observation regime. This mirrors practice in climate / weather foundation models (ClimaX-, Pangu-style), where one backbone is adapted to specific tasks or lead times via fine-tuning. Retraining only the controller is substantially cheaper than retraining the full ARDM.
>
> * **Compared to per-instance optimization.**
>   Instead of solving a new optimization problem for *each* assimilation window (as in diffusion posterior sampling or variational control methods like NDTM), our controller amortizes the inverse mapping and can then be reused for arbitrarily many assimilation cycles.
>
> * **Practicality.**
>   In operational DA, observation regimes (sensor layout, cadence) typically change on much slower timescales (months) than the assimilation cycle (hours). Training a small controller per relatively stable regime is therefore realistic, especially when a neural surrogate is already in use.
>
> ---
>
> ### 4. Tilted objective, KL term, and Eq. 8–10
>
> We have **reorganized the method section**:
>
> * The “tilted” problem (Eq. 3 in the revision) is presented as the *ideal* entropy-regularized control objective: minimize assimilation cost plus KL between the controlled path distribution and the prior ARDM path distribution.
> * We set β=0 for stability, but the controller acts only over short preview windows with small step size γ, which implicitly constrains how far each transition can deviate from the prior and thus the KL. Empirically, we do not observe reward collapse or pathological drifts: energy spectra and other diagnostics remain close to those of the pretrained ARDM.
>
> ---
>
> ### 5. Hyperparameter tuning
>
> * We use the authors’ public code and recommended hyperparameters, with a small grid search only where our regimes differ.
> * The comparison is in fact favorable to the baselines: they operate with very large context windows whereas our ARDM prior uses single step transition. Despite this narrower context, CADA achieves lower DA errors and better long-horizon stability.
>
> ---
>
> ### 6. Methodological clarity
>
> We have rewritten the following:
>
> * The **revised Sec. 2** now clearly separates:
>   (i) the variational objective and DA problem;
>   (ii) the unguided and guided ARDM dynamics;
>   (iii) controller training on short preview windows; and
>   (iv) the sliding-window inference scheme.
> * We updated the pseudocode so that the controller’s inputs are clearly specified as **ingesting (z_{t+1}^{(s+1)})** too, and **line 6 in Alg 1 clarifies how Tweedie-based arrival costs are accumulated**.

---

### Author Response · Authors · 2025-11-23
**Author response to reviewers**

We thank the reviewers for their constructive feedback. Reviewers agree that the paper addresses an important problem at the intersection of diffusion models and data assimilation, that the high-level narrative is clear, and that the approach is novel. Reviewer yjRG explicitly considers the work above the acceptance threshold and indicated they would raise their score to 8 once additional baselines and clarifications were incorporated. Building on this, the revised version substantially **strengthens the empirical evidence and improves the exposition**, while keeping the core method unchanged.

### 1. Stronger empirical scope and baselines

Some concerns were about empirical breadth and practical relevance. We addressed these with four concrete additions:

1. **ERA5-based NWP-style experiment.**
   We add a compact but realistic ERA5 case study (500 hPa winds and temperature over North America), using the same controller architecture as in the PDE setups. CADA outperforms the diffusion-based DA baselines in this setting, demonstrating that the method transfers to an NWP-style surrogate without changes to the core algorithm.

2. **Classical DA baselines.**
   In addition to diffusion-based DA methods, we now include EnKF, 3DVar, and 4DVar on the Kolmogorov setup, using identical observation operators and cadences and the pretrained ARDM as the forecast model. These results show that CADA maintains lower error and better long-horizon stability.

3. **Irregular observation networks.**
   To address concerns about regular grids, we add an experiment with randomized spatial masks and jittered observation times. Using the same controller architecture, CADA remains stable and performs best across methods, indicating that the framework naturally accommodates irregular observation patterns.

4. **Runtime and computational cost.**
   We report wall-clock time in the main text. Because all optimization of the controller is done offline and inference is a single ARDM rollout plus a lightweight controller call, CADA is significantly faster than reconstruction-guided and joint-score baselines, which require repeated gradient/score evaluations or inner optimization per window.

### 2. Clearer objective, training pipeline, and method flow

Reviewers also asked for a crisper formal problem statement and a more explicit training description. In the revised manuscript we:

* Introduce a **Problem Statement** subsection that defines the pretrained autoregressive prior ( $\mathcal{Q}$ ), the **tilted posterior** ( $\mathcal{P}^* \propto \mathcal{Q}\exp(-\sum_\tau \Phi(x_\tau;y_\tau)/\beta) $), and the associated variational objective that balances observation fit and KL to the prior.
* Make the **training loss explicit** for anchored preview windows, showing how the parametric path distribution ( $\mathcal{P}_\psi$ ) induced by the controlled kernel is optimized using arrival-time costs.
* Reorganize the method into a **top–down pipeline**: (i) problem + tilted objective, (ii) unguided and guided ARDM dynamics, (iii) learning the control policy, and (iv) preview windows and sliding-window inference, supported by Algorithms 1–3.

These changes directly answer the requests from JdQ9, z35e, and hrmH for a clearer formulation and make the training / inference pipeline easy to follow and reproduce.

### 3. Positioning relative to guidance, test-time training, and DA practice

Finally, some comments asked whether CADA is simply another form of guidance or test-time training. We sharpen this positioning in the revised manuscript:

* **No test-time optimization.** All learning of the controller ($u_\psi$) is done once, offline. At inference, both the ARDM and ($u_\psi$) are frozen; we perform a single causal rollout with small control injections. This contrasts with test-time training and reconstruction-guided methods, and is confirmed empirically by our faster wall-clock times and a TTO ablation that is slower and less stable.
* **Controller as a “likelihood head”.** We explicitly frame the ARDM as a reusable “foundation” forecast prior and the controller as a lightweight per-regime assimilation head, matching how climate / weather foundation models are adapted in practice.
* **Limitations and scaling.** We add a brief limitations discussion: the main cost lies in training the ARDM prior (shared with existing surrogates), controllers are per observation regime but small, and scaling to full global NWP will likely require localized or meta-initialized controllers, natural directions for future work.

---

In summary, the revised manuscript retains the strengths highlighted by the reviewers (novelty, clear high-level narrative, and strong performance on challenging DA regimes) and directly addresses their main requests: **ERA5 validation, classical DA baselines, irregular observation experiments, explicit computational analysis, and a clearer training pipeline.**

---

### Meta-Review · Area_Chair_Xg5K · 2026-01-08

**Summary:**

The paper proposes "Control-Augmented Data Assimilation" (CADA), an amortized control framework for autoregressive diffusion models designed to bypass the computational overhead of test-time optimization in data assimilation (DA). While reviewers acknowledged the novelty of the amortized approach and the demonstrated inference speedups, the submission is recommended for rejection. This decision is primarily driven by the following three critical concerns:


Marginalization of Domain Standards (Reviewers JdQ9, yjRG): Despite claiming to address the core challenges of Data Assimilation, the authors relegated the field's "gold standard" baselines (EnKF, 3D/4DVar) to the appendix. This placement undermines the confidence in the method's competitiveness. To demonstrate genuine value, the method must be benchmarked against these established mathematical frameworks in the main results, rather than solely comparing within a limited scope of AI-based models.

Practical Inflexibility (Reviewer z35e): The framework exhibits a critical rigidity: changes to observation locations or frequencies require retraining the controller network. This is a significant practical flaw compared to traditional methods (such as 4DVar) or optimization-based baselines, which can natively handle variable observation conditions without retraining.

Limited Experimental Scale (Reviewers z35e, hrmH, yjRG): Although an ERA5 experiment was included, the setup remains restricted in scale and complexity. It serves merely as a "proof-of-concept" and fails to convincingly demonstrate that the method can scale to the demands of real-world global weather prediction systems.

**Reviewer Concerns:**

see above

**Reviewer Scores:**

Reviewer z35e: 6 (Marginal Accept) -> 6 (Marginal Accept) . Reasoning: While the reviewer appreciated the narrative, the fundamental concern regarding the practical inflexibility of the "controller-per-regime" design remains unresolved. The added ERA5 experiment was "compact" and did not fully satisfy the requirement for real-world relevance.

Reviewer JdQ9: 0 (Strong Reject) -> 2 (Reject). Reasoning: The authors addressed the significant presentation issues and provided code. However, the reviewer's critical request for a comparison with standard methods (3D/4DVar) was only partially met by placing them in the appendix for a single dataset, which is insufficient to justify acceptance.

Reviewer yjRG: 6 (Marginal Accept) -> 6 (Marginal Accept). Reasoning: The reviewer stated they would raise their score if classical baselines were added. However, since these crucial results were marginalized to the appendix rather than integrated into the main evaluation, the validation remains less convincing than required for a higher score.

Reviewer hrmH: 2 (Reject) -> 4 (Reject). Reasoning: The authors clarified the reviewer's misunderstanding regarding test-time fine-tuning. However, the reviewer's skepticism regarding the motivation and the "toy" scale of the experiments compared to real-world DA scenarios persists.

---

### Decision · Program_Chairs · 2026-01-26

Reject